

# Hydration Status and Diurnal Trophic Interactions Shape Microbial Community Function in Desert Biocrusts

Minsu Kim[1] and Dani Or[1]

[1]Department of Environmental Systems Sciences (USYS), ETH Zürich, 8092 Zürich, Switzerland

*Correspondence to:* Minsu Kim (minsu.kim@usys.ethz.ch)

**Abstract.** Biological soil crusts (biocrusts) are self-organised thin assemblies of microbes, lichens and mosses ubiquitous in arid regions serving as important ecological and biogeochemical hotspots. Biocrust ecological function is intricately shaped by strong gradients of water, light, oxygen and dynamics in abundance and spatial organisation of the microbial community within a few millimetres of the soil surface. We report a mechanistic model that links biophysical and chemical processes that

shape the functioning of biocrust representative microbial community that interacts trophically and responds dynamically to cycles of hydration, light, and temperature. The model captures key features of observed microbial activity and distribution (during early stages of biocrust establishment) and associated dynamics of biogeochemical fluxes. The study offers new insights into the highly dynamic and localised processes that shape biocrust functioning and elude quantification based on averaged representation of such delicate and globally important ecological assembly.

## 1 Introduction

Large tracks of arid lands are often covered by thin biological soil crusts (hereafter, biocrusts) that, in the absence of significant vegetation cover, play an important role in the arid ecosystems. Biocrusts serve as biodiversity "hotspots" (Belnap et al., 2016) and act as ecosystem engineers to promote rehabilitation of eroded soils in arid lands (Bowker, 2007). The photoautotrophs inhabiting biocrusts support rich and diverse food webs and provide the main source of organic carbon covering for over 70 %

of arid land surface area (about 30 % of all terrestrial surfaces; Belnap and Lange (2002); Mager (2010)). Biocrust microbial activity produces extracellular organic exudates that alter the immediate environment by supporting a stable structure, altering water retention and transport properties of the biocrusts (Mazor et al., 1996; Belnap and Lange, 2002; Belnap, 2003). The resulting modification of local hydrological processes such as infiltration-runoff and water storage (Chamizo et al., 2012), enhances the capability of other organisms to cope with water scarcity (Chamizo et al., 2016). Furthermore, this water-regulating

function of biocrusts also protects soil surface against wind and water erosion (Belnap and Gillette, 1998; Warren, 2001).

    Evidence suggests that biocrusts are locally and globally important component of the ecosystem in terms of biogeochemical fluxes; arid land biocrusts affect global cycles of carbon and nitrogen (Weber et al., 2016b). Biocrusts regulate carbon dioxide efflux through soil by fixing $\sim 0.6$ Pg of carbon per year, which is about 9 % of the net primary productivity of this ecosystem (Sancho et al., 2016; Elbert et al., 2012). Their contribution to nitrogen fixation from the atmosphere is even more

significant, evaluated as about 26 Tg per year, corresponding to about 40 % of the global terrestrial biological nitrogen fixa-



tion (Elbert et al., 2012; Ciais et al., 2014). Although biocrust contribution to terrestrial nitrogen fixation is considerably high, arid land ecosystems remain largely nitrogen-limited due to the substantial losses of nitrogen gas caused by abiotic (temperature, pH) and biotic (nitrification, denitrification) processes (Peterjohn and Schlesinger, 1990; McCalley and Sparks, 2009). The global emission of reactive nitrogen (such as NO, HONO) from biocrusts has been estimated at about 20 % of the globally

emitted reactive nitrogen compounds from natural soils (Weber et al., 2015).

Biocrusts are sensitive and highly vulnerable systems to anthropogenic and natural disturbances, leading to the erosion of the invaluable microbial community (Kuske et al., 2012). Natural recovery of biocrusts is a slow process (multidecades) (Weber et al., 2016a), and the recovery rates may vary widely depending on precipitation, soil texture, or carbon content (Weber et al., 2016a). The recovery stage follows a general successional pattern beginning with surface soil colonisation by mobile

cyanobacteria such as *Microcoleus vaginatus* (Büdel et al., 2009; Zaady et al., 2010). The settlement of photoautotrophic organisms is followed by other phototrophic, heterotrophic and chemoautotrophic microorganisms, algae, and fungi, etc. At the later stages of biocrust formation, cyanobacteria, the main primary producers, are replaced by other photoautotrophic organisms such as mosses or lichens. Most established biocrusts consist of microscopic and macroscopic organisms within the top few centimetres of the soil surface (e.g., around 5 mm thick for cyanobacterial crusts and up to 5 cm thick for moss crusts).

A typical biocrust community consists of hundreds of species, representing different levels of trophic interactions that enable an entire arid land ecosystem to function systematically (Bowker et al., 2010a, b).

The composition and structure of a biocrust are determined by several environmental factors. At a local scale, soil properties such as texture, nutrient level, and pH, are the main determinants (Bowker et al., 2016). At a global or regional scale, the characteristics of a biocrust community differ with climatic regions (from cold to warm deserts), soil type, and crust age since last

disturbance (Garcia-Pichel et al., 2013; Bowker et al., 2016). Regional climatic variables such as the amount of precipitation or the potential evapotranspiration influences the biomass of cyanobacteria and other photoautotrophs, as a consequence, define the community composition (Isichei, 1990; Hagemann et al., 2015; Barnard et al., 2015). Studies have shown that cyanobacterial crust distribution and their activity are highly correlated with periods between rain events and soil water availability rather than precipitation amount of a single rain event (Lange, 2003; Cable and Huxman, 2004; Büdel et al., 2009). Thus, the response

of microbial activity to wetting events, such as precipitation, is a crucial factor in the ecology of biocrusts.

Notwithstanding the importance of these ecosystems, quantitative studies using mathematical or computational approaches to link the complex biological, physical, and chemical processes that underlie this sensitive ecosystem have been scarce. Many field and laboratory studies have focused on statistical analyses of the results to deduce impacts of various environmental factors on observed biocrust response (Barger et al., 2006; Grote et al., 2010; Bowker et al., 2010a; Castillo-Monroy et al., 2011;

Maestre et al., 2013). This study reports a mechanistic model for the early stages of biocrust formation and key biophysical and chemical processes. We construct a representation of hydrological processes within a biocrust and trophic interactions among key members of biocrust microbial community. The model includes a detailed account of the physical domain available for microbial life (simple rough surfaces) and the consequences of different hydration conditions on connectivity and transport of nutrients, gas, temperature, and light. The model also considers dynamic chemical processes. The key ingredient in

biocrust functioning is the highly dynamic and spatially self-organising microbial community. For simplicity, we considered





four microbial groups: photoautotrophs (primarily cyanobacteria), aerobic heterotrophs, anaerobic heterotrophs (denitrifiers), and chemoautotrophs (nitrifiers) (Garcia-Pichel and Belnap, 2002; Johnson et al., 2005, 2007; Abed et al., 2010, 2013; da Rocha et al., 2015), to consider their role in carbon and nitrogen cycling.

The organisation of this paper is as follows: We first introduce the key physical and chemical processes in the mechanistic
model. Next, the biochemical feedback of microbial activity and its spatial organisation is investigated. The results of this model are compared with data obtained from laboratory experiments. Finally, we provide new insights into the ecological functions of unsaturated soil structures in established biocrusts in arid regions.

## 2   A mechanistic model of desert biocrusts

The study was motivated by interest in biocrusts as a model ecosystem supporting multi-species microbial community that
interact at a limited spatial extent under large environmental gradients (Bowker et al., 2014). We employ individual based modelling of microbial processes in the presence of sharp environmental gradients in resources and conditions. The model addresses first the physical domain and its dynamics characteristics that vary with hydration conditions. Chemical and biological processes are then introduced into the physical domain (associated primarily with the aqueous phase and its distribution).

### 2.1   The biocrust physical domain

We use a modified rough surface patch model (Šťovíček et al., 2017; Kim and Or, 2016) to represent the top millimetres to centimetres of soil where most biocrusts develop (see Fig. 1a). For the physical domain, we consider a vertical cross-section of a biocrust that considers rough soil grains and the gas pathways between grains (described in 2-D but in a simplified fashion including 3-D features in a spatial element). Geometrically-explicit features of the rough surface (pyramid-shaped depressions of different sizes) are averaged according to a probability distribution of the pore sizes (representing roughness
decorating soil grains). For the size distribution, we assign three parameters, local porosity $\phi$, surface roughness porosity $\Phi$, and fractal dimension $D$ (Šťovíček et al., 2017; Kim and Or, 2016). This abstract representation of the physical domain permits physically-based calculation of the amount of water held within the rough surface for given matric potential (our standard hydration metric). The water films then determine the diffusion rates and pathways of nutrients, microbial dispersion rates and ranges, connectivity, and the complementary spaces for gas diffusion. While the representation of microbial life is
assigned to the two-dimensional rough surface, the inference of gas phase within the cross-section is applied into the model by considering two rough surfaces facing each other (see Fig. 1a). This approach allows us to extend the surface model to the vertical cross-section model and to include gas diffusion and mass transfer between liquid and gas phases as a function of matric potential by using effective film thickness (Fig. 1c) without the complexity of 3-D modelling of the pore space in the biocrust (Kim and Or, 2016; Šťovíček et al., 2017).
We represent a section through the biocrust by a spatially distributed soil properties assigning key parameters $\{\phi, \Phi, D\}$ to individual patches (representing soil grains or small aggregates). This domain represents a strong heterogeneity of soil structure including regions with low or high porosities. For simplicity, we assume in this study that the matric potential is constant for





the entire biocrust. Hence, the water distribution in the model biocrust including phase connectivity and related properties were predetermined for a simulation. Although evaporation or drainage processes following (rare) rainfall events can generate hydraulic gradients across the biocrust, these effects can be neglected given the small domain size ($< 10\,\mathrm{mm}$).

## 2.2 Environmental boundary conditions

The model includes three essential environmental variables that shape microbial community in desert soil: water, light, and temperature. For simplicity, we prescribe the hydration status of the biocrust, this status determines the configurations of the liquid and gas phases (that, in turn, determines the respective diffusion coefficients). The extension to dynamic hydration conditions is relatively simple considering infiltration, redistribution and soil evaporation (these processes are functions of the crust properties and are representable analytically or numerically). Temperature and light are applied as time-dependent

boundary conditions to the top of the crust domain to mimic diel cycles.

### 2.2.1 Light irradiance on the surface

Light determines the photosynthetic activity of the phototrophs (e.g., cyanobacteria) within the biocrust (Berner and Evenari, 1978; Davies et al., 2013). To represent light penetration and diurnal day-night cycle, we express irradiance, $I(z;t)$, as a function of depth $z$ and time $t$,

$$I(z;t) = \begin{cases} \frac{I_0}{2}\left(1 - \cos(\omega t + \phi_I)\right)e^{-z/\delta_p} & \text{day} \\ 0 & \text{night} \end{cases} \qquad (1)$$

where $I_0$ is maximum irradiance (at midday on the biocrust surface). Incident irradiance at the surface is given by the period $P \equiv \frac{2\pi}{\omega}$ (24 hours) and $\phi_I = 0$ (with $t = 0$ at sunrise, 6AM). $\delta_p$ is the light characteristic penetration depth. The values of $I_0$ and $\delta_p$ regulate the activity and spatial location for an optimised growth of phototrophs in the model. The sinusoidal function with $I_0$ and $\omega$ can be changed with respect to the specific location of the biocrust and the season of the year (and even the

local aspect and slope of the surface). The value of $\delta_p$ varies from about $10^{-4}$ to $10^{-3}$ m depending on the amount of mineral or soil texture (grain size distribution) (Garcia-Pichel and Bebout, 1996). In this work, we chose a maximum irradiance of $I_0 = 500\,\mathrm{\mu mol.m^{-2}.s^{-1}}$ corresponding to the light intensity of overcast sky (assuming that a biocrust shows activity when it is wet, i.e. during rainy days). The calculations consider a constant light penetration depth of $0.2\,\mathrm{mm}$ (Garcia-Pichel and Belnap, 1996), and only vertical penetration is considered in the current work. The resulting distribution of irradiance in the model is

depicted in Fig. 1b.

### 2.2.2 Temperature dynamics

The profile of soil temperature varies with time and space following a periodic function coupled with light incidence. We consider ambient temperature as a sinusoidal function for the surface boundary condition (Phillips et al., 2011):

$$T(z = 0, t) = \bar{T} + A_0 \sin\left(\omega t + \phi_T\right) \qquad (2)$$





where $\bar{T}$ is the average temperature on surface and $A_0$ is the diurnal amplitude. The period $P \equiv \frac{2\pi}{\omega}$ is assumed to be one day and the phase is set to be 0, so the maximum temperature corresponds to the maximum of light intensity (midday). Considering a homogenous domain with uniform hydration status, an analytical solution for a 1D heat equation with sinusoidal temperature boundary condition (Equation (2)) yields a dynamic description of diurnal soil temperature profile (similar solution is obtained for seasonal profiles).

$$T(z,t) = \bar{T} + A_0 e^{-\frac{z}{d}} \sin\left(\omega t + \phi_T - \frac{z}{d}\right) \tag{3}$$

where $d$ is a characteristic damping depth of the domain given by $d = \sqrt{\frac{P\alpha}{\pi}}$ where $\alpha \equiv \frac{\lambda}{c_v}$ is thermal diffusivity (for details, see Supplementary Materials: Text 1). Thermal diffusivity is a function of hydration conditions. i.e., conductivity increases with wetness. Soil temperature distribution over depth during a diurnal cycle for wet and dry conditions is illustrated Fig. 1b.

## 2.3 Biocrust biogeochemical processes: mass transfer/inorganic C and N partitioning

The chemical environment of soil is strongly influenced by its microbial activity (especially at the main biocrust region). For example, the pH of biocrust is altered diurnally due to microbial respiration (release of protons and bicarbonates) and photosynthesis, with $CO_2$ removal significantly modifying pore water pH. These, in turn, affect nutrient availability and mobility, $CO_2$ dissolution rates, and solubilisation of soil minerals (Belnap et al., 2002). The model includes certain essential chemical processes; diffusion, gas-liquid phase partitioning, and acid-base dissociation, that affect microbial activity within typical biocrusts.

### 2.3.1 Gas diffusion with the biocrust

Unsaturated conditions dominate microbial life in desert biocrusts and support unhindered gas diffusion most of the time. Gas diffusion coefficient is in the order of $10^{-6}$ m$^2$.s$^{-1}$, which is about $10^4$ times greater than that in aqueous phase. The largely aerated biocrust, the partial pressures of soil gas near surface equilibrate with the atmospheric level almost instantly (it takes a few seconds to aerate soil at depth of a few milimetres). In contrast, when the soil surface becomes especially wet, the aqueous phase configuration may temporarily hinder gas diffusion and delay such instantaneous partial pressure equilibration. Thus, an understanding of water configuration within the domain is necessary. In unsaturated soils, water is held on the rough soil surface due to the capillary force (given by Young-Laplace equation) and absorbed waterfilm (van der Waals force). The abstract model (Fig. 1a) provides a means for calculating the proportion of water held at the given hydration conditions from preassigned soil properties. This yields the degree of saturation after normailsation with the volume of void space and local gas/water content (proportion of gas/water) of each spatial element (a patch). We used these local properties for gas phase invasion probability and local diffusion coefficients. When a patch at location $r$ is connected to the atmosphere (invasion percolation), constant boundary conditions at the gas phase are assigned in respect of atmospheric mixing ratio of each gaseous element instead of resolving gas diffusion at near surface (assuming instant equilibration).



### 2.3.2  Mass transfer between gas and liquid

The mass transfer rate across the gas-liquid interface can be determined by using Fick's law and the film model.

$$\frac{\Delta C^l}{\Delta t} = -\frac{A_{lv}}{d_{\text{tot}}}\left(\frac{D_g}{1-\Theta} - \frac{D_l}{\Theta}\right)(C^l - C^g) \equiv -k_{l\leftrightarrow g}(C^l - C^g), \tag{4}$$

where $C^l$ and $C^g$ are substrate concentration in liquid and gas phases, respectively; $A_{lv}$ m$^2$.m$^{-3}$ is the specific liquid-vapour
interfacial area; $d_{\text{tot}}$ is the effective thickness of soil pore space; $D_l$ and $D_g$ are diffusion coefficients for liquid and gas phases;
$\Theta$ is the degree of saturation; and $k_{l\leftrightarrow g}$ is the net mass transfer rate across the interface, which is a function of hydration
conditions.

The proposed model allows us to calculate the specific liquid-vapor interfacial area and the effective thickness of void
space (Kim and Or, 2016). For instance, the model estimates that $A_{lv} \approx 10^5$ m$^2$.m$^{-3}$ and $d_{\text{tot}} \approx 10^{-5}$ m at $\Theta = 0.5$ (half
saturation). These values are consistent with other studies that have used a preassigned water retention curve (Zand-Parsa and
Sepaskhah, 2004). Considering that gas diffusion coefficient is in the order of $10^{-6}$ m$^2$.s$^{-1}$, the net mass transfer rate between
two phases is $\sim 10^4$ s$^{-1}$ in aerated soils. This implies that the concentration at liquid phase also equilibrates almost instantly to
the concentration at gas phase. Even when the soil is nearly saturated, the rate is $\sim 1-10$ s$^{-1}$ ($\Theta \to 1, \theta \approx \theta_s$) (For comparison,
studies on waste water treatment used the rate of $\approx 6.9 \times 10^{-5}$ s$^{-1}$ (Buhr and Miller, 1983; Yang, 2011).). Thus, mass transfer
between gas and liquid in unsaturated soils is assumed to be rapid, and the concentration at each phase is always at equilibrium
following Henry's law.

$$C^{l^*} = H_{cc}(T)C^{g^*} \tag{5}$$

where $H_{cc}(T)$ is a dimensionless Henry's constant at temperature $T$; $H_{cc} = H_{cc}^S e^{-\frac{\Delta_{\text{soln}}H}{R}\left(\frac{1}{T} - \frac{1}{T^S}\right)}$ where $\Delta_{\text{soln}}H$ is the enthalpy
of solution, $R$ is the gas constant, and $S$ refers to the standard condition ($T^S = 298.15$ K) (Sander, 1999).

### 2.3.3  Dissociation of chemical substances

To evaluate available $CO_2$ in soil-biocrust water, one must consider the open-system behaviours of $CO_2$ and the different
species of dissolved inorganic carbon (DIC), carbonic acid (H2CO3), biocarbonate ($HCO_3^-$), and carbonate (CO3$^{2-}$). The
relative amounts of such DIC species can be determined by the concentration of protons, pH of the solution. Considering that
most desert soils are alkaline (Bresler et al., 2012) (implying the predominant DIC species to be bicarbonate), the determination
of the amount of dissolved $CO_2$ in soil is essential to the growth and functioning of autotrophs and to distinguishing between
abiotic and biotic processes for $CO_2$ efflux estimation. Assuming that other chemical species are inert, the model includes the
following geochemical reactions focusing on carbon and nitrogen dynamics in soil:

$$H_2O \rightleftharpoons OH^- + H^+ \tag{R1}$$

$$CO_2(aq) + H_2O \rightleftharpoons HCO_3^- + H^+ \tag{R2}$$



$$HCO_3^- \rightleftharpoons CO_3^{2} - + H^+ \tag{R3}$$

$$NH_3(aq) + H^+ \rightleftharpoons NH_4^+ \tag{R4}$$

$$HNO_2(aq) \rightleftharpoons NO_2^- + H^+ \tag{R5}$$

$$CaCO_3^0(aq) \rightleftharpoons Ca^{2+} + CO_3^{2-} \tag{R6}$$

Some mathematical models have introduced pH estimation for systems with phototrophs under light-dark cycles, such as
algal ponds (Buhr and Miller, 1983; Yang, 2011; Gomez et al., 2014) and phototrophic biofilms (Wolf et al., 2007). The algal
pond models invoke solution equilibrium and charge neutrality and employ differential algebraic equations to estimate pH,
while the phototrophic biofilm models consider acid-base reactions with rate equations by proposing near-equilibrium kinetics.
The unsaturated conditions in desert biocrusts with large air-liquid interfacial areas and high mass transfer rates require a
special treatment. We adopted a similar approach of kinetics with charge balance (Wolf et al., 2007). In addition, the range
of geochemical reactions were extended by including nitrous acid (HONO) and nitrous oxide ($N_2O$) to investigate nitrogen-
related gas emissions from biocrusts. In this work, calcium is considered enabling evaluation of biogenic precipitation of
calcium carbonate in biocrust formation. All the kinetics are based on the local concentration of each substrate in pore water
with an assumption of water activity 1.

The equilibrium gas phase concentrations of $O_2$, $CO_2$, $NH_3$, $N_2O$, and HONO are considered in the model according to
Henry's law. Considered reactions for gas and liquid phase partitioning and precipitation are listed below:

$$CO_2(aq) \rightleftharpoons CO_2(g) \tag{R7}$$

$$NH_3(aq) \rightleftharpoons NH_3(g) \text{ (volatilisation)} \tag{R8}$$

$$HNO_2(aq) \rightleftharpoons HONO(g) \tag{R9}$$

$$N_2O(aq) \rightleftharpoons N_2O(g) \tag{R10}$$



$$CaCO_3^0 \rightleftharpoons CaCO_3(s) \text{ (precipitation)}. \tag{R11}$$

Values and detailed kinetic equations used in the model are summarised in Supplementary Materials: Text 2.

## 2.4 Microbial community in desert biocrust ecosystem

Advances in molecular taxonomic techniques and DNA sequencing have greatly expanded our knowledge on microbial community structure and diversity in biocrusts. These data generally delineate the interplay between multi-level trophic interactions (Bowker et al., 2011; da Rocha et al., 2015; Pepe-Ranney et al., 2016) and surrounding environmental conditions (Caruso et al., 2011). Biocrusts host a complex community of diverse autotrophs and heterotrophs (hundreds of species including about 20 generic or subgeneric taxa of cyanobacteria) (Bowker et al., 2010a, b; Garcia-Pichel et al., 2013). Considering biocrusts

as independent and self-sufficient ecosystems, the intrinsic diversity found in this system should not come as a surprise. The incorporation of natural microbial diversity found in biocrusts is beyond the present capabilities of most models. Hence we opted for a representation of the main microbial actors for modelling of associated biogeochemical cycles in a cyanobacterial crust.

### 2.4.1 Microbial community and trophic interactions

Four functional microbial groups are represented in the *in silico* microbial model of a desert biocrust: photoautotrophs, aerobic heterotrophs, anaerobic heterotrophs (denitrifiers, strictly anaerobes using $NO_3^-$ as a terminal electron acceptor), and chemoautotrophs (nitrifiers). These groups are chosen to elucidate the interlinked functionality of C/N cycling in a biocrust microbial community. Thus, we considered the following substrates in the soil solution that support microbial activity: oxygen ($O_2$), dissolved inorganic carbon, DIC, ($CO_2$/ $HCO_3^-$), ammonium ($NH_4^+$), oxidised nitrogen species ($NO_3^-$, $NO_2^-$), and organic

carbon ($CH_2O$, as an elementary form of polyglucose). Here, phototrophically produced $CH_2O$ is assumed to be the primary carbon source available, which can be transformed into extracelluar polymeric substances (EPS) depending on environmental conditions. Other chemical species, $Ca^{2+}$, $CO_3^{2-}$, $N_2O$, $NH_3$, and $HNO_2$, are included for the study of chemical reactions but are not directly utilised by these microbial species.

The four microbial groups interact based on prescribed stoichiometric relations (see Supplementary Materials: Table S5 in

Text 4). These stoichiometric relations require the photoautotrophs to be classified into four subgroups (Wolf et al., 2007), using one inorganic carbon source and one inorganic nitrogen source during photosynthesis (i.e. $CO_2$+$NH_4^+$, $HCO_3^-$+$NH_4^+$, $CO_2$+$NO_3^-$, and $HCO_3^-$+$NO_3^-$). Aerobic heterotrophs use $CH_2O$ as an electron donor, $O_2$ as an electron acceptor, and $NH_4^+$ as a nitrogen source. Anaerobic heterotrophs (denitrifiers) use $CH_2O$ as an electron donor, $NO_3^-$ as an electron acceptor as well as a nitrogen source. As obligate anaerobes, their growth is inhibited by the presence of oxygen. Chemoautotrophs are described

in two subgroups, considering two oxidation processes, firstly, of ammonia to nitrite by ammonia oxidising bacteria (AOB) and, secondly, of nitrite to nitrate by nitrite-oxidising bacteria (NOB).




By using Monod-type kinetics with limiting substrates, the growth rate of species $i$, with a limiting factor $j$ can be written as;

$$\mu_i = \mu_{\max,i} \min[f_i^1, f_i^2, \cdots, f_i^j],\tag{6}$$

where $\mu_{\max,i}$ is the maximum growth rate of species $i$ and Monod factors are of two types, $f_i^j = \frac{C_j}{K_{S,i}^j + C_j}$ (when nutrient $j$

is a substrate for the growth) or $f_i^j = \frac{K_{i,i}^j}{K_{i,i}^j + C_j}$ (when nutrient $j$ is an inhibitor of growth) (For details, see Fig. 2 and see Supplementary Materials: Table S7 in Text 4).

The proposed model describes the various roles of phototrophs (i.e., cyanobacteria) within its growth dynamics by including the activity switch between photosynthesis and dark respiration, regulation of C/N ratio via $N_2$ fixation by heterocysts, and production of EPS. By means of adapting their growth stoichiometry to the local environment, phototrophs in the model control the primary productivity of the entire system depending on the time of the day (photosynthesis, dark respiration), nutrient availability (unbalanced C/N ratio), and hydration conditions (EPS production). A detailed description of the activity of phototrophs is provided in Supplementary Materials: Text 3.

To evaluate stoichiometries of heterotrophs and nitrifiers, microbial metabolic reactions are explicitly considered using the MBT-Tool (Metabolism based on Thermodynamics) (Araujo et al., 2016). Details of stoichiometry for microbial growth in the model can be found in Supplementary Materials: Table S5 in Text 4. A graphical summary of microbial growth and trophic interactions is given in Fig. 2

### 2.4.2 Temperature-dependent microbial growth

Desert environments are often characterised by large diurnal temperature fluctuations (especially in hot deserts), which influence microbial activity. To consider these thermal effects, a temperature-dependent growth model using Arrhenius equation is included in the model. Although temperature adaptation and growth adjustments may vary among microbial species, we opted for a simple representation where all species are assumed to follow the same optimal temperature. The maximal growth rate for a cell at temperature $T$ is scaled as follows (Schoolfield et al., 1981):

$$f_T = \left[ \frac{\frac{T}{T^S} e^{\frac{\Delta H^S}{R}\left(\frac{1}{T^S} - \frac{1}{T}\right)}}{1 + e^{\frac{\Delta H_L}{R}\left(\frac{1}{T_L} - \frac{1}{T}\right)} + e^{\frac{\Delta H_H}{R}\left(\frac{1}{T_H} - \frac{1}{T}\right)}} \right],\tag{7}$$

where $T^S$ is reference temperature ($25°C = 298K$) and $\Delta H^S$ (cal.mol$^{-1}$) is activation enthalpy of the reaction. In this model, two inactivation regimes are considered, one of low temperature, denoted by $L$, and other of high temperature, denoted by $H$. Parameters included for enthalpies and inactivation regimes are given in Supplementary Materials: Table S9 in Text 5.

### 2.4.3 pH feedback

Our model considers the spatial and temporal variations in pH values that could locally affect microbial activity. Unlike the narrow range of high pH regulating the activity of autotrophs (often limited by dissolved organic carbon), the activity of heterotrophs in the presence of dark respirations likely lowers pH when other substrates are not limited. Furthermore, nitrate




accumulation can result in acidification of the soil domain when denitrification is absent. Considering that high acidity and alkalinity profoundly affects microbial growth through substrate binding and catalyse reactions, the feedback of microbial growth to local pH change is included in the model. The microbial feedback on biocrust pH can vary based on types of enzymes, number of ionisable groups, and organisms under consideration. In this work, a non-competitive inhibition model in

a form of Monod function is employed (Tan et al., 1998):

$$f_{pH} = \frac{K_{pH}}{K_{pH} + [H]} \tag{8}$$

where $K_{pH}$ is inhibition constant that deactivates microbial growth at very low pH (in this work, microbial activity ceases at pH below 5, $K_{pH} = 10^{-5}$ [M]). Usually, $K_{pH}$ is a function of binding energy although it is implemented as a constant in our model for simplicity. Unlike other pH-dependent growth models, the inhibition term for hydroxyl ions is not included since

the resulting high pH will regulate DIC and its partitioning will limit microbial growth without any inhibition terms (lack of protons for activity).

### 2.4.4   Microbial growth rates

A key objective of our model is to determine the spatial organisation of microbial community based on local gradients in conditions and resources. Several biocrust physico-chemical properties and environmental conditions determine the microbial

growth rate following the diel cycles of light, temperature, and feedback of pH. As a result, the growth rate of individual cell $i$, Equation (17), is explicitly expressed as

$$\mu_i(\boldsymbol{r},t) = \mu_{\max,i} f_T(\boldsymbol{r},t) f_{pH}(\boldsymbol{r},t) \min[f_i^1(\boldsymbol{r},t), f_i^2(\boldsymbol{r},t), \cdots]. \tag{9}$$

Here, substrates are described within their minimum function (mass limitation of electron donors/acceptors) unlike pH and temperature correction terms. We assume that $f_T$ indicates the optimal temperature of enzymes, and $f_{pH}$ indicates the costs of

osmosis of protons; therefore, they act on the maximum growth rate directly.

### 2.5   Microbial EPS production

The importance of EPS for microbial life in natural environments has been discussed in many review articles (Or et al., 2007; Flemming and Wingender, 2010; More et al., 2014). Especially, in arid or semi-arid environments, the role of EPS secreted by cyanobacteria is crucial for microbial communities surviving within (and below) biocrusts (De Philippis and Vincenzini,

1998; Pereira et al., 2009; Mager and Thomas, 2011; Rossi et al., 2012; Colica et al., 2014; Rossi and De Philippis, 2015). The synthesis of EPS contributes to the stability of soil structure and hydrated microenvironments in soil, making it a key ingredient of biocrust formation. EPS also functions as a nutrient storage by immobilising nutrients (dust trapping or glycosidic bonds) and as a protective shield from adverse environments, such as UV radiation, antibiotic substances, and invasion of viruses. In this work, we focus on two key aspects of EPS in biocrusts: modification of diffusion process of substrates and its role as a

nutrient reservoir (increase in soil C) (Or et al., 2007; Pereira et al., 2009; Mager and Thomas, 2011). The complete range of




EPS effects on soil hydrology, such as swelling of hydrated gel, owing to its chemical composition and physical structure, are not considered in this study.

### 2.5.1 EPS production and transport properties

EPS production by cyanobacteria in drylands varies with soil type, climatic conditions, hydration status and other resources (Hu et al., 2002). Estimation of production rates and amounts remain challenging. It is generally accepted that EPS synthesis in cyanobacterial soil crusts is affected by changes in moisture availability and nitrogen level (Mager and Thomas, 2011). We thus coupled photosynthesis and $N_2$ fixation in the biocrust model. This approach allows to compute the net production of carbohydrates using dynamic stoichiometry. A certain proportion of carbohydrate produced is assumed to be transformed into EPS depending on the local hydration conditions (for details, see Supplementary Materials: Text 3.3).

The fraction of EPS produced from photosynthetically fixed carbon is defined by the biding of extracellular carbohydrate residues to the polymeric matrix. The binding probability is written as a function of EPS concentration $C_{EPS}$ and saturation degree $\Theta$ in the model:

$$f_p(C, \Theta) = \frac{1}{e^{-\frac{C_{EPS} - C_{EPS}^*}{C_{EPS}^* \Theta}} + 1} \tag{10}$$

where $C_{EPS}^*$ is gelation point for EPS as a polymeric substance. The function describes that residual carbohydrate will not bind to the polymeric substances anymore as soon as EPS is in a form of a weak gel (reaching $C_{EPS}^*$). The degree of polymer binding is regulated by the saturation degree. For example, when the domain is wet, EPS hydrolysis will lower the binding probability of newly produced residual carbohydates.

Many studies have suggested different physical models to describe the diffusion coefficient in EPS (Masaro and Zhu, 1999). For our biocrust model, we adopted a simple diffusion model in gels proposed by (Phillies, 1987).

$$D = D_0 e^{-\alpha_d c^\nu} \tag{11}$$

where $\alpha_d$ and $\nu$ are scaling parameters that differ from substance to substance. It is shown that $\alpha_d$ depends on the diffusant's molecular weight (in g.L$^{-1}$) and $\nu \sim 0.5$ for a high-molecular-weight diffusant (macromolecules). Diffusion of carbohydrates and EPS is governed by this equation in the model.

### 2.6 Diffusion reaction equation at the biocrust scale

Microbial activity and resource consumption are expressed as a set of diffusion reaction equations within the biocrust domain.

$$\frac{\partial C_j(\boldsymbol{r}, t)}{\partial t} = \nabla \cdot (D_j(\boldsymbol{r}, t) \nabla C_j(\boldsymbol{r}, t)) - \frac{1}{V_w(\boldsymbol{r}, t)} \sum_{i=1}^{N(\boldsymbol{r})} \frac{\mu_i(\boldsymbol{r})}{Y_{\text{net } j}^i} b_i(t) + S_j(\boldsymbol{r}, t), \tag{12}$$

where $C_j(\boldsymbol{r}, t)$ is the local concentration of substrate $j$, $D_j(\boldsymbol{r}, t)$ is the local diffusion coefficient (including modification by EPS), and $V_w(\boldsymbol{r}, t)$ is the amount of water in a given patch at position $\boldsymbol{r}$ and time $t$. The second term on the right-hand side is the reaction term to calculate the total substrate consumption/production in the patch. $N(\boldsymbol{r})$ is the total number of individual



cells at $\boldsymbol{r}$, $Y_{\mathrm{net}\,j}^{\,i}$ is the net yield of species $i$ on substrate $j$, $b_i(t)$ is the biomass, and $\mu_i(\boldsymbol{r})$ is the growth rate described in Equation (20). The last term $S_j(\boldsymbol{r},t)$ is the source or sink term of substrate $j$ with respect to the mass transfer between gas and liquid phases and charge compensation from principles of solution equilibrium and charge neutrality. These chemical processes are very fast compared to microbial reaction and diffusion processes. Thus we implemented these terms as dynamic

boundary conditions (keeping gaseous element solubility and local charge neutrality during one time step). For individual cells, the growth dynamics is written as:

$$\frac{db_i(t)}{dt} = \left[\mu_i(\boldsymbol{r},t) - m_i\right] b_i(t) \tag{13}$$

where $\mu_i(\boldsymbol{r},t)$ is growth rate, from Equation (20), and $m_i$ is maintenance rate of cell $i$. Cell growth, division, locomotion, and death are described using the Individual Based Modelling (Kim and Or, 2016; Kreft et al., 1998).

## 2.7   Evaluation of the proposed mechanistic Desert Biocrust Model (DBM)

A pioneering study on microbial community within desert biocrusts (Garcia-Pichel et al., 2003) has suggested a vertical stratification of microbial community members where abundance (biomass) and composition (functional groups) vary with depth of the biocrust. Observations by (Garcia-Pichel et al., 2003) demonstrated that the abundance of bacterial community members as a result of vertical gradients in physicochemical conditions such as light, oxygen, pH, and other nutrients. Vertical

profiles of $N_2$ fixation and potential $NH_4^+$ oxidation rates (Johnson et al., 2005), chemical profiles (total ammonium, nitrate) of soil solutions within active biocrusts (Johnson et al., 2007), and profiling of oxygen concentration after wetting (Abed et al., 2014) have been investigated as well. Recently, the effect of physical conditions on cyanobacterial activity was examined using X-ray microtomography (Raanan et al., 2015). These experimental data on microprofiles within biocrusts can be used for a comparison between measurements and numerical simulations of chemical/biological components within saturated crusts.

In contrast with the generally dry state of biocrust, most of the detailed studies reported above were conducted using saturated biocrusts (a state that rarely occurs in the field). Data on unsaturated biocrusts are hindered due to the technical difficulty in using microsensors (Pedersen et al., 2015) and molecular analysis of microbial activity (Carini et al., 2016). Consequently, we are left with the undesired option of using detailed data from saturated biocrusts for model evaluation. The primary aim of this study is to establish confidence in the DBM for these rare conditions and extend the predictions to the more common case of

unsaturated biocrusts.

The DBM was evaluated with respect to diurnal dynamics and results are compared to experimental studies that measured certain traits (e.g., gaseous efflux) such as the studies of (Thomas et al., 2008; Rajeev et al., 2013; Darrouzet-Nardi et al., 2015; Weber et al., 2015). In this work, we focus on carbon dioxide efflux under fully saturated conditions (Rajeev et al., 2013). Although the gaseous fluxes are usually considered direct indicators of microbial activity, quantitatively speaking, these

macroscopic measures emerged from all possible biological, chemical, and physical interactions combined.




## 2.8 Physical domain and boundary conditions for nutrients

For a prescribed matric potential (constant hydration conditions), the corresponding water film thickness, aqueous habitat connectivity, diffusion properties, and specific surface area are obtained locally at patch scale (about $100\,\mu m$) from preassigned surface properties and local porosity of each patch (a spatial element that determines local patch property). We selected parameters that mimic the property of loamy sand ($\overline{\phi} = 0.4$, $\overline{\Phi} = 0.6$, $D = 2.65$). By applying $m \times n$ patches, the domain describes a thin strip of a biocrust with periodic boundary conditions in the horizontal direction.

For boundary conditions of chemical substances, gaseous elements and dissolved elements are treated differently. Oxygen, carbon dioxide, ammonia, nitrous oxide, and nitrous acid in gas phase are assigned based on the atmospheric composition from literature (see Supplementary Materials: Table S1 in Text 2). The mixing ratios of atmospheric components are kept constant at the top of the domain during simulations assuming zero diffusive boundary layer and maximised gas exchange between atmosphere and biocrusts. These gaseous compounds are transferred to liquid phase by their own solubility based on Henry's law (Sander, 1999). DIC, ammonia, and nitrous acid are partitioned with the principle of local charge neutrality at obtained pH values.

Model evaluation is based on the following ingredients: We first present steady state distribution of geochemical variables within the biocrust domain. Next, we present quasi-steady distribution of microbial functional groups within the biocrust under field capacity (relatively wet conditions). We then compare model results for saturated conditions where sample experimental data are available.

## 3 Results

### 3.1 Steady state of geochemical traits within the biocrust (no biological activity)

The abiotic exchanges that affect local distributions of geochemical environments and traits are evaluated first. A steady state of chemical domain is calculated in the absence of biological activity. We consider a biocrust following wetting at field capacity (corresponding to water saturation of 0.6 for the entire domain) assuming that this condition describes wetted crusts after drainage (in contrast to a fully saturated crust with saturation degree 1). We focus on traits such as diffusion, gas-liquid partitioning, and acid-base calculation) without microbial activity. The spatial variations in phase distributions within the simulation domain (vertical cross-section of biocrust) and related attributes are depicted in Fig. 3. The results suggest that these relative wet conditions may disrupt gas phase connectivity to the atmosphere. Gas diffusion through the biocrust is determined by the connectedness of gas phase according to percolation theory. For certain values of local gas content (below 0.2), the gas phase becomes disconnected, affecting $O_2$ distribution. This implies that gas volumes not connected to the atmosphere may exist in isolated pockets within the soil domain. Thus, the local concentration of dissolved oxygen varies according to this atmospheric source and spatial heterogeneity (Fig. 3c). This also shows a correlation between gas phase configuration and spatial heterogeneity of pore water pH; the higher the local gas content, the lower the pH values (activity of protons). This indicates that a higher mass transfer rate from gas to aqueous phase yields acidity since the dissolution of $CO_2$ is very fast in



unsaturated soils (large surface areas and thin water films). On the other hand, patches with high water content and limited gas phase penetration show higher pH (around 8-9) as the model mimics alkaline soils with high cation content (about 10 $\mu g.g^{-1}$ calcium and same amount of other nonreactive cation as shown in (Johnson et al., 2005)). This implies that volume-averaged pH may not be representative of local soil pore water/waterfilm pH in unsaturated soil, thereby affecting microbial activity

locally and giving rise to processes not definable by average values.

### 3.2   Microbial activity effects on the biocrust chemical environment

The four microbial groups are introduced into the simulation domain (representing a cross-section in desert biocrust) and allowed the system to stabilise under diurnal cycles. Phototrophs were initially inoculated in the domain in an exponentially decaying manner over the biocrust depth to reflect a natural organisation under light penetration, while other groups were inoc-

ulated uniformly in the domain. Only phototrophs were inoculated differently to reduce the computational time as phototrophs only thrives up to the depth where light penetrates. This well-mixed inoculation pattern assures that the spatial organisation of microbial populations within the crust was not affected by initial conditions. The initial population sizes were the same for all functional groups, about 4000 cells, for the entire domain. After about five consecutive days (diurnal cycles), the total population/spatial distribution of microbial groups reached a quasi-steady state.

Noticeable changes in the resulting chemical environments occurred due to microbial activities even though physical environments and hydration conditions were assumed to be constant (held at relatively wet conditions corresponding to field capacity). Fig. 4 depicts four spatially distributed chemical attributes, namely dissolved oxygen, pH, total ammonia nitrogen, and nitrate, for midday (top panel) and midnight (bottom panel). The chemical profiles delineate the diurnal cycles of microbial activity across the soil domain. For instance, the alkalisation of top crust (2 mm) was clearly shown together with the produc-

tion of ammonium. This implies phototrophic activity removes inorganic carbon to fix carbon as well as produces ammonium to fix $N_2$ using heterocysts. However, the oxygen profile was relatively stable compared to other chemical substances although photosynthesis and dark respiration could introduce changes in the local concentration of dissolved oxygen. This is due to the unsaturated conditions on the top crust, where gas transfer rates override the net reaction rate of oxygen within the profile. In addition, the nitrate profile also exhibits spatio-temporal diurnal cycles with the tendency of cumulation below 4-5 mm,

implying that inhibited denitrification under unsaturated conditions. In general, regardless of differences among various chemical species and diurnal cycles, the strong spatial heterogeneity was still significant within the domain shaped by gas-liquid configuration.

### 3.3   Vertical stratification of microbial functional groups

The dynamics of the biocrust chemical environments are not only due to general microbial activity, but specifically due to inter-

nal trophic interactions within the biocrust (due to different substrate use by microbial groups). The activities and interactions among biocrust microbiota under two distinctive phases are depicted in Fig. 5: (1) during daytime with active photosynthesis (A-D), (2) during nighttime with dark respiration (E-H). Here, the activity is quantified as a product of local growth rate, $\mu_i$,



and biomass, $b_i$, of cells with a unit of $\mu g_{cell} \cdot g_{soil}^{-1} \cdot h^{-1}$:

$$A_i(\boldsymbol{r},t) = \mu_i(\boldsymbol{r},t)b_i(\boldsymbol{r},t). \tag{14}$$

We chose this activity measure instead of population size because this measure indicates the active pathways for the upregulation of functional genes (i.e., spatial distribution of gene activity). From this activity distribution, it is possible to calculate the
rates of microbial processes, such as carbon fixation, ammonia oxidation, denitrification, etc, by simply multiplying the yields from the stoichiometry of each species.

   Results show emergence of vertical stratification of each microbial process within the thin biocrust (10 mm). The spatial pattern is driven by trophic interactions among groups, by the chemical environments, and resource gradient since the non-phototrophic cells were uniformly inoculated over the entire domain. We note that, although activity and growth rates were in
diel cycles, the spatial pattern become relatively steady and migration is not observed although cell motility is enabled (each population reached its local carrying capacity). The patterns can be analysed as follows: The phototrophs as primary producers (green in figure) perform intense photosynthesis at the biocrust's top following the distribution of light. The produced oxygen and carbohydrates combined with $N_2$ fixation benefit aerobic heterotrophs (yellow in figure) that exhibit high activity 2 mm below the surface. This strong cooperation between phototrophs and aerobic heterotrophs support high population on the top
of the crust. Although a close proximity (mixing) between phototrophs and aerobes is expected, their activities are segregated due to the strong alkalisation during photosynthesis and intense competition over ammonium with AOB (marked in dark blue). A weak activity of anaerobic bacteria is also found together with aerobes at a similar depth due to the need of organic carbon for their activity. Local anoxic conditions support their growth in certain regions due to the consumption of oxygen by other organisms, heterotrophs, and nitrifiers. Below 3 mm, anaerobic activity is not found because the oxygen consumption by
aerobic organisms is too low to create local anoxic conditions. Chemoautotrophs appear sparse over the depth and AOB and NOB (light blue) stay in proxy as they are in a mutualistic relation. AOB shows high activity within 2 mm during daytime, benefiting from ammonium fixed by heterocysts of phototrophs and inorganic carbon produced by heterotrophs. Its growth is mainly limited by inorganic carbon used during photosynthesis. The activity of NOB is also high at the top crust due to nitrite production by AOB.

Generally, during daytime, the activity of phototrophs enhances other microbial activity by fixing inorganic carbon and nitrogen (Fig. 5d). During nighttime, phototrophs switch their activity to dark respiration. Dark respiration by phototrophs drives an intense competition for organic carbon and ammonium among individuals at the top of the domain. As the input of fixed carbon and nitrogen is absent, the depletion of ammonium at the top crust lowers the activity of most organisms (Fig. 4g). However, NOB shows slightly higher activity during night at below 3 mm, suggesting that, during daytime, they are
outcompeted by other organisms owing to their high yield and low growth rate.

### 3.4 Fully saturated biocrusts: comparing model predictions with observations

Despite the focus of the desert biocrust model (DBM) on unsaturated conditions in desert systems, we had to rely on definitive experimental data from saturated biocrusts to evaluate details of model performance (Garcia-Pichel and Belnap, 1996; Johnson



et al., 2007; Abed et al., 2013; Rajeev et al., 2013; Raanan et al., 2015). The simulation domain was saturated by simply applying near zero matric potential and filling up all surface pores with water. Using the fully saturated domain with stable microbial community distribution, the model biocrust was then exposed to diurnal cycles of radiation and temperature.

The spatial distribution of microbial activity within a fully saturated biocrust is given in Fig. 6. Ten independent simulations
were averaged to obtain the possible distribution of microbial processes. The potential activity of anaerobes peaks below 3 mm (in contrast to other aerobic organisms) due to the formation of an anoxic region (Supplementary Materials: Fig. S4 in Text 6). At the top, microbial distribution is clearly stratified as phototrophs-nitrifiers-aerobic heterotrophs-denitrifiers. Unlike unsaturated biocrusts, the vertical stratification is accentuated largely because of a strong oxygen gradient profile driven by photosynthesis.

The spatio-temporal behaviour of oxygen and pH profiles predicted by the model are compared with available dataset in Fig.7. Simulation results are in quantitatively agreement with reported data from experiments on various types of cyanobacterial crusts (e.g. light crusts and dark crusts) from several locations (Garcia-Pichel and Belnap, 1996; Johnson et al., 2007; Abed et al., 2013; Rajeev et al., 2013; Raanan et al., 2015). A common finding with respect to the oxygen profile is its supersaturation within a top few millimetres and the formation of an anoxic region below. While the model was able to capture the dynamics
of dissolved oxygen, pH dynamics showed large deviations between model and data, especially during nighttime. Chemical environments of other substrates during daytime and nighttime are given in Supplementary Materials: Fig. S4 in Text 6.

### 3.5   Diurnal cycles of gaseous efflux from saturated biocrusts

In addition to comparing processes within the crust (Fig. 6, 7), we simulated gas efflux from the saturated biocrust and compared with the measurements of (Rajeev et al., 2013). Fig. 8 depicts the efflux of three gas compounds of carbon and nitrogen, namely
$CO_2$, $NH_3$, and $N_2O$. We represent uptake by negative gas efflux and positive sign for emissions. The diel cycles of $CO_2$ efflux are plotted together with experimental data tracking the net carbon exchange between the biocrust and the atmosphere (Fig. 8a). Within the biocrust, carbon fixation and respiration occur simultaneously; the net $CO_2$ efflux indicates a balance between respiration (release) and photosynthesis (uptake). Simulation results are in a qualitative agreement with experimental data, except the steep transitions after sunrise and gradual changes after sunset that are not captured properly. We attribute
this to the simplified model (using Monod functions) of the onset of photosynthesis and dark respiration. Next we evaluate the daily patterns of ammonia volatilisation to represent nitrogen abiotic losses. The results in Fig. 8b show that ammonia volatilisation occurs mainly during daytime as the top of the biocrust turns alkaline (pH above 10). The total ammonia loss due to volatilisation was estimated to be about 500 $nmol.m^{-2}day^{-1}$, similar to reported values, $540 \sim 1000$ $nmol.m^{-2}day^{-1}$, from intact biocrusts in Colorado plateau (Evans and Johansen, 1999; Barger et al., 2016). We then evaluate $N_2O$ release
from the biocrust (indicative of denitrification), the results in Fig. 8c show that immediately after wetting $N_2O$ flux is high. We attribute this rapid release to accumulation of $NO_3^-$ during unsaturated conditions. After 2 days, nitrate is exhausted and denitrification relies on the activity of NOB. Finally, we also considered the potential release of $NO_2^-$ from the soil solution in the form of nitrous acid HONO. Results show, however, no such release in agreement with the observations of (Weber et al., 2015).



## 4 Discussion

### 4.1 Spatial and temporal variations in local pH within unsaturated biocrusts

Soil pH has been recognised as a significant predictor of microbial community composition and diversity (Fierer and Jackson, 2006; Lauber et al., 2009). Furthermore, for alkaline or saline soils (typical desert soils), abiotic contributions to gaseous efflux
may account for up to 40 % of total $CO_2$ emissions (Ma et al., 2013). Thus, to separate biotic and abiotic contributions for gaseous efflux, reliable estimates of pH are needed. The proposed desert biocrust model (DBM) offers a distinct advantage in this respect, namely the localised (pore scale) representation of pH that integrates physicochemical interactions and microbial activity. Simulated pH profile dynamics within wet biocrusts presented above (Fig. 7) have confirmed that the activity of photoautotrophs alters local pH by depleting DIC during a diel cycle (consistent with observations).

Results by the DBM suggest strong spatial variations of local pH within the unsaturated biocrust although the overall (spatially averaged) soil pH indicates an alkaline soil (Fig. 3). In practice, however, the spatial distribution of local soil pH is difficult to measure and often requires the use of microelectrodes (Pedersen et al., 2015). Moreover, it has been argued that the use of microsensors is limited to near-saturated soils (McIntyre, 1966). The modelled spatial variations in local acidity are consistent with uptake kinetics of nitrous acid in the gas phase on a wetted wall film (Hirokawa et al., 2008). Model results
suggest that pH in thin water films may be lower than in the bulk liquid due to the resistance of mass transfer from the gas to the bulk liquid phase (we use the term "bulk" to represent large water-filled pores within the biocrust). As liquid surface on the wall corresponded to acidity at thin water film in the model, this result may support model predictions and the importance of soil water configuration in shaping local pH within unsaturated soils.

    The strong correlation between soil moisture retention and soil pH and their role in defining the microbial community struc-
ture (Lauber et al., 2009) might be attributed to local pH distribution in unsaturated soil. We speculate that the high abundance of *Acidobacteria* (at phylum level), known to grow well at acidic culture (pH 3.5-6.5) as aerobic heterotrophs (Pankratov et al., 2008), in most soils (Jones et al., 2009; Lauber et al., 2009) might offer another evidence of the importance of localised acidity in unsaturated soils. We note that such acidity related phylum was also found in biocrust communities (Steven et al., 2013).

### 4.2 Microbial community stratification within biocrusts

Spatial segregation along vertical gradients is a well-known feature of microbial communities in aquatic biofilms, microbial mats, and endolithic communities (Schramm et al., 2000; Paerl et al., 2000). Similar to the Winogradsky column, these microbial stratifications are driven by the distribution of electron acceptors/donors. Since the most favourable electron acceptor for aerobic organisms is oxygen, the low solubility of oxygen and the limited diffusion of dissolved oxygen play a pivotal role in the emergence of spatial stratification. The stratification within biocrusts is also observed in terms of biomass of oxy-
genic phototrophs, aerobic copiotrophs (Garcia-Pichel et al., 2003), and community composition analysis based on 16S rRNA sequencing (Steven et al., 2013). The simulated results of our biocrust model agree with observations exhibiting vertical stratifications of the biocrust community (Fig. 5 and 6).



The DBM captures the key physicochemical conditions essential for vertical stratifications. The steep gradient of oxygen on top of the fully saturated biocrust (Fig. 7, Supplementary Materials: Fig. S4 in Text 6) is caused by limited mass transfer from the atmosphere and rapid consumption of oxygen. During nighttime, depletion of oxygen (below few millimetres) is expected naturally because of limited amount of oxygen input. The oxygen produced by phototrophs during daytime is imme-

diately depleted by aerobic organisms in the domain. The creation of supersaturation closer to the surface also indicates slower diffusion than net production/consumption of oxygen. Experiments on biocrusts immersed in water indicated effervescing of (presumably) oxygen at surface (Rajeev et al., 2013). This proves that the net production of oxygen is higher than diffusion of dissolved oxygen. Clearly, such formation of anoxic region within the crust benefits anaerobic activity (Fig. 6).

The vertical segregation of different microbial groups also indicates diffusion of organic carbon and nitrogen diffusing from

the photoautotrophs and become available to other microbial members, especially, stratification among aerobic organisms. The dominance of nitrite oxidising bacteria (NOB) at the top 2 mm is largely due to ammonia volatilisation. The alkalisation of the top crust during daytime increases ammonia volatilisation, which is not beneficial for aerobic heterotrophs and ammonia oxidising bacteria (AOB). Therefore, their activity retreats deeper to around 2 mm, allowing NOB to appear at the top surface. Below the location of NOB, we find AOB and aerobic heterotrophs. Although the model has a simple assumption on microbial

groups utilising various substrates in a specific trophic landscape assumed for this study (Fig. 5d, h), a similar pattern of segregation is expected within real biocrusts in field.

For aerated unsaturated biocrusts, the results in Fig. 4a and e show that the high oxygen transfer rate to soil water overrides net reaction, and thus a strong gradient of oxygen is not observed in unsaturated cases (Supplementary Materials: Fig. S4A in Text 6). Therefore, the aqueous phase configuration within unsaturated biocrusts (also possibly extending to general unsaturated

soils) shapes microbial activity unlike in aquatic, microbial mats and similar saturated systems. This may also explain the large abundance of aerobic anoxygenic phototrophic bacteria within certain biocrusts (Csotonyi et al., 2010).

The biocrust community exhibits highly dynamic and complex trophic interactions, such as commensalism surrounding organic carbon utilisation between phototrophs and heterotrophs, competition over nitrogen sources between aerobic heterotrophs and AOB, cooperation between NOB and anaerobic denitrifiers, etc. Temporally, diel patterns of trophic interactions (orches-

trated by phototrophs) drive the shift in activity distribution of microbial activity as it has been shown from Namib desert soil (Gunnigle et al., 2017). Spatially, these complex trophic interactions take place within thin biocrusts and yield emergent spatial distribution of microbial groups as depicted in Fig. 5. The remarkable concentration of such interactions within a few mm and the stratification of the activities of the various functional groups highlight the ecological sophistication and versatility of such fine tuned desert ecosystems. Remarkably, opportunistic life forms are harboured within such biocrusts for example,

the presence of anaerobic heterotrophs that are present at low numbers suggesting presence of local anoxic conditions even under mild unsaturated conditions (Ebrahimi and Or, 2015) and their rapid response to episodic wetting events (Šťovíček et al., 2017).





### 4.3 Gaseous efflux from desert biocrusts

Motivated by availability of definitive data, the DBM was applied to simulating diurnal changes in gas efflux from saturated biocrusts. The results were in good agreement with measured $CO_2$ efflux (Fig. 8). The model represents diurnal cycles of other gas fluxes that may be sensitive to pH such as ammonia volatilisation, HONO emission, etc. Details of the geochemical

environment shed light on the important role of local conditions (pH) on soil/biocrust microbial activity. For example, the activity of AOB in alkaline soils can be suppressed during daytime on the top crust as strong alkalisation leads to a loss of nitrogen compound. On the other hand, NOB in acidic soils should experience the opposite, as the soil becomes more acidic, HONO emission would lead to nitrogen loss.

To realistically describe microbial life within unsaturated biocrusts or dry soils, the inclusion of gas phase interactions is

necessary. Most experiments on biocrusts were conducted under saturated conditions (presumably to induce significant and measurable response), however, these responses occur during narrow climatic windows with high precipitation (Garcia-Pichel and Belnap, 1996, 2002; Johnson et al., 2007). Although we have shown gaseous efflux from saturated soils to compare with experimental results, the DBM is capable of quantifying gaseous efflux from unsaturated biocrusts by tracking gas and water distribution.

### 4.4 Assumptions and limitations of the desert biocrust model (DBM)

The proposed DBM makes numerous simplifications pertaining to the life and functions of a complex microbial community in biocrusts in arid and semi-arid regions. Regarding the key physical processes, we built a physical domain that contains small subregions represented as patches. A patch is a subsection within a small vertical cross-section in the biocrust that represents soil surfaces with different properties that retain water films and transport nutrients and gas. This enables consideration of

spatial heterogeneity within a vertical two-dimensional cross-section across a biocrust, however, lateral variations in biocrust properties in space are not considered here.

Key geochemical processes that are dominant in desert soils (and biocrusts) are considered in this model. For simplicity, we consider calcium as a buffer together with other non-diffusing background cations (assuming uniformly distributed non-reactive cations as a setpoint of pH). The effects of saline soil (also a common property of desert soils) on dissociation constants

and its influence on soil pH are not considered. We also did not include the effect of EPS (as organic matter) on the top of the biocrust. The role of EPS as a gate for matter flux on desert soil surface, interaction between pH alteration by microbial activity, and changes in physical properties of soil (relation between EPS swelling ratio and pH) can be the next goal for a mechanistic model of biocrusts. Other important aspects regarding chemical processes include modifying the diffusion equation. In the current model, the possibility of electrokinetic flow is not included. A more detailed description of electromigration can be

included by modifying the diffusion equation for ionic particles by using the Nernst Planck equation. However, as the input of carbon dioxide to the thin water film is faster than the aqueous diffusion of ionic particles, the occurrence of local pH variation owing to the configuration of gas phase is still expected in unsaturated soil.



By far, the most simplified component in this model is the biological one related to microbial processes. The DBM represents a system containing an astonishing level of diversity with a small number of microbial functional groups. The interactions among these community members are regulated by simple stoichiometric relations that control microbial growth. Monod parameters are mostly taken from models for activated sludge (a system far removed from life in desert biocrusts) (Henze, 2000). Considering that a desert is a water-, carbon-, and nitrogen-limited system with abiotic stresses, the values of these parameters are likely to be different from those governing life in sludge systems. We note however, that the proposed Monod growth parameters are affected by local environmental conditions, such as temperature, pH, and substrate concentrations. Yet, an understanding of half-saturation constants and ratios between growth rates among different microbial groups would be necessary for establishing quantitative predictions by the DBM for real systems.

The members of the biocrust consortia were selected focusing on C and N cycling and characteristics of arid environments. Recently, the role of heterotrophic diazotrophs, anaerobic ammonium oxidisers, and nitrate-reducing bacteria within biocrusts has been studied. We are aware that including these members might alter some of expected rates that we presented in this study. Comparisons between crust models with their presence and absence can be one of the future applications. Furthermore, as the model describes an hydrated porous medium, the fully saturated domain is easily applicable to describe the microbial community of sediments or microbial mats. However, when it comes to modelling such systems, other groups such as anaerobic phototrophs, sulfate/iron-reducing bacteria, or methanogens might need to be considered together with the proposed community of C/N cycling. This might be beneficial for a mechanistic understanding of the biogeochemistry of such systems.

The DBM can be further used to predict the gaseous efflux dynamics of wetting-drying cycles and C and N turnover rates during hydration events. As hydration events in arid and semi-arid areas are scarce, a mechanistic understanding of biocrust response to hydration would benefit to estimate its contribution to global biogeochemical cycles. For instance, high N loss via $NO_3^-$ leaching, $NH_3$ volatilisation, and HONO emissions can be investigated with respect to N cycling in such environment. Furthermore, short term perturbations of hydration conditions on biocrusts can be also another application of the model, such as short wet-up cycles or rapid evaporation at high temperatures.

## 5 Summary and conclusions

In this study we develop a mechanistic model of desert biocrust microbial community under strong vertical resource gradients prevailing in surfaces of arid landscapes. The desert biocrust model (DBM) combines a detailed account of soil hydration for different soil properties, individual based description of microbial life, and chemical processes that affect the trophic interactions among the microbial groups as an ecologically functioning unit. Although simplified (as much as possible) it elucidates the role of soil structure in shaping gaseous/aqueous diffusion and substrate fluxes at the atmosphere-soil interface crucial for microbial activity occurring therein.

Model results show the distribution and composition of microbial functional groups over vertical gradients of light, temperature, and substrates across a model biocrust. Furthermore, geochemical and physical processes of mass transfer at the gas-liquid interfacial area in soil matrix and kinetics for inorganic carbon and nitrogen fractionation underline the importance of modelling




unsaturated soil that significantly deviates from other environments such as aquatic systems or saturated soils. Especially, the modified chemical environment displays the feedback of microbial activity from photosynthesis to $CO_2$ efflux from biocrusts. The local pH of soil water as a cumulative measure of local ionic species concentrations determines the availability of inorganic carbon and nitrogen or other minerals for microorganisms by controlling the solubility of chemical compounds and their

5 degree of protonation. Although the model does not include individual differences of optimal pH for microbial activity, its results based on acid-base equilibrium predict the spatially and temporally organised activity of all functional groups. This self organisation indicates one of the reasons why biocrusts can host high abundance and diversity of microorganisms even under very harsh conditions like deserts. The DBM provides a means for systematic and climatic-driven evaluation of the critical role of microorganisms in desert ecosystems. The model offers predictive capabilities (within the limitations of the assumptions)

10 for biocrust responses to climate change and their contribution to large scale carbon and nitrogen cycles.

*Code and data availability.* Additional data and materials are available online. The codes for the DBM are available upon request from corresponding author (minsu.kim@usys.ethz.ch).

*Author contributions.* DO conceived the research. MK and DO conducted the analyses and wrote the manuscript

*Competing interests.* The authors declare that they have no competing financial interests.

15 *Acknowledgements.* The authors thank Daniel Baumann for IT supports. This work was supported by the European Research Council (ERC) Advanced Grant (320499-SoilLife) and the SystemsX.ch (2013-158:MicroScapesX project).





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





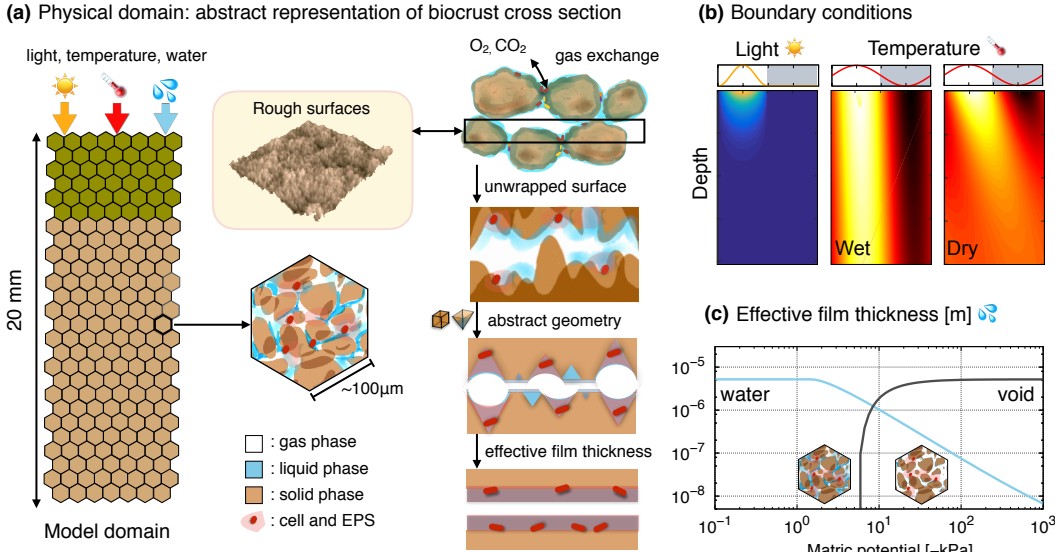

**Figure 1. A schematic of physical domain and environmental conditions of the desert biocrust model (DBM).** (a) A cross-section of the physical domain of the modelled biocrust. The domain comprises hexagonal patches with different physical properties (mimicking soil pores and rough surfaces). The rough surface is simplified with abstract geometries to calculate effective film thicknesses of the surface at patch scale. To represent interference of liquid phase with respect to gas diffusion, we consider two rough soil surfaces (cross-sections) facing each other as described previously (Kim and Or, 2016; Šťovíček et al., 2017). (b) Spatio-temporal variations of light intensity and temperature as boundary conditions (wet and dry) during a diurnal cycle. Surface boundary conditions of temperature changes in accordance with light irradiance during the same period of a day. Unlike light penetration, temperature profile depends on hydration conditions (thermal diffusivity is controlled by the matric potential). Under wetter conditions, thermal diffusivity is higher. (c) Effective thicknesses of water film and void space are determined as a function of matric potential.

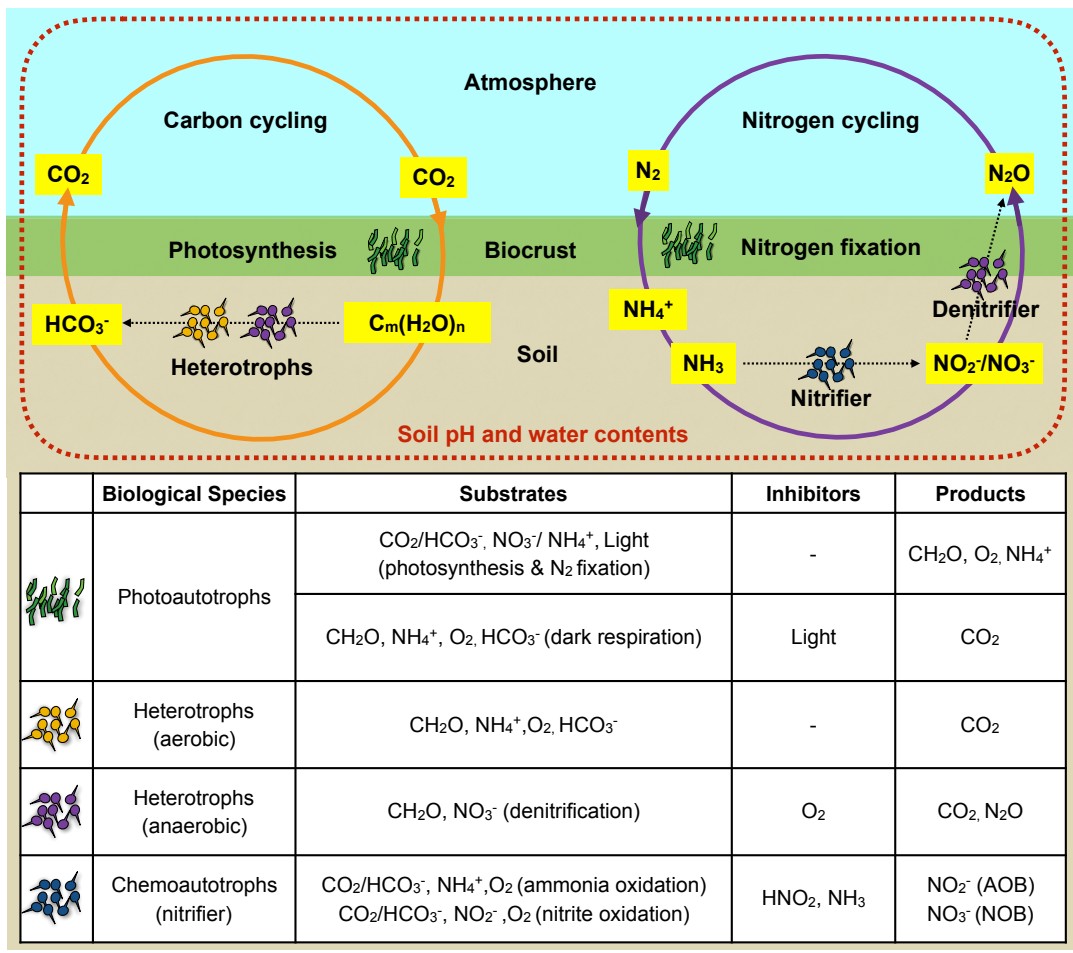

**Figure 2. Key microbial functional groups and biogeochemical interactions within the desert biocrust model.** The biocrust is considered an ecological unit where four groups of biological species to describe carbon and nitrogen cycling. The introduced chemical and biological species in conjunction with the chemical processes determine the dynamics of local pH of soil pore water and gaseous efflux at the top of the domain. The growth rate of each species is determined from Equation (17). For details of stoichiometry, rate expressions, and Monod parameters, see Supplementary Materials: Text 4.



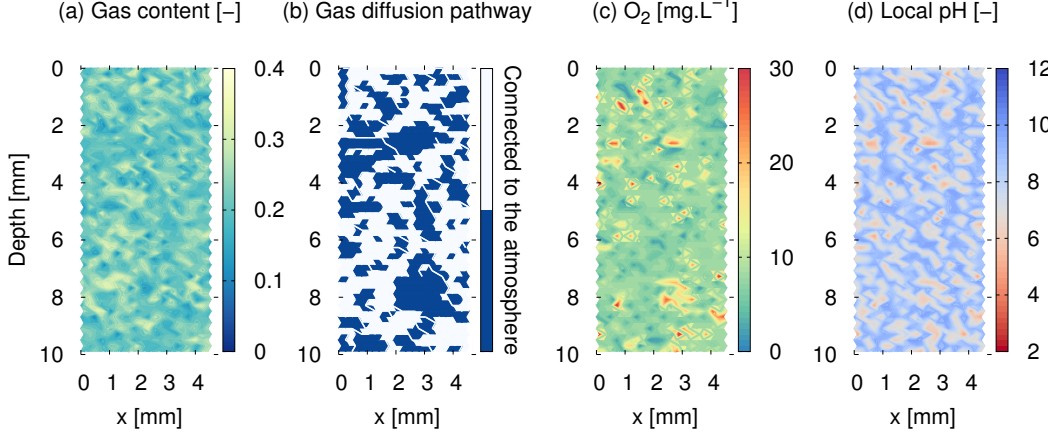

**Figure 3. Aqueous phase distribution affects diffusion pathways and geochemical conditions (no biological activity)** A typical simulation result of a steady-state soil biocrust (up to 10 mm depth) when biological activities are absent at standard ambient temperature (T = 25 °C). (a) A preassigned soil structure determines the local gas content and configuration of water at field capacity (the aqueous phase is complementary in these pore spaces). (b) The unsaturated soil permits gas phase to penetrate over the biocrust depth along pathways (marked in blue) not blocked by the aqueous phase (marked in blue). The process is described by invasion percolation in this study. When the gas phase is connected to the atmosphere, partial pressures of gaseous compounds equilibrate to the atmospheric level as boundary conditions. Gas and liquid phase configurations determine the distribution of chemical species in liquid phase, (c) The distribution of dissolved oxygen concentration, and (d) localised soil pore water pH.



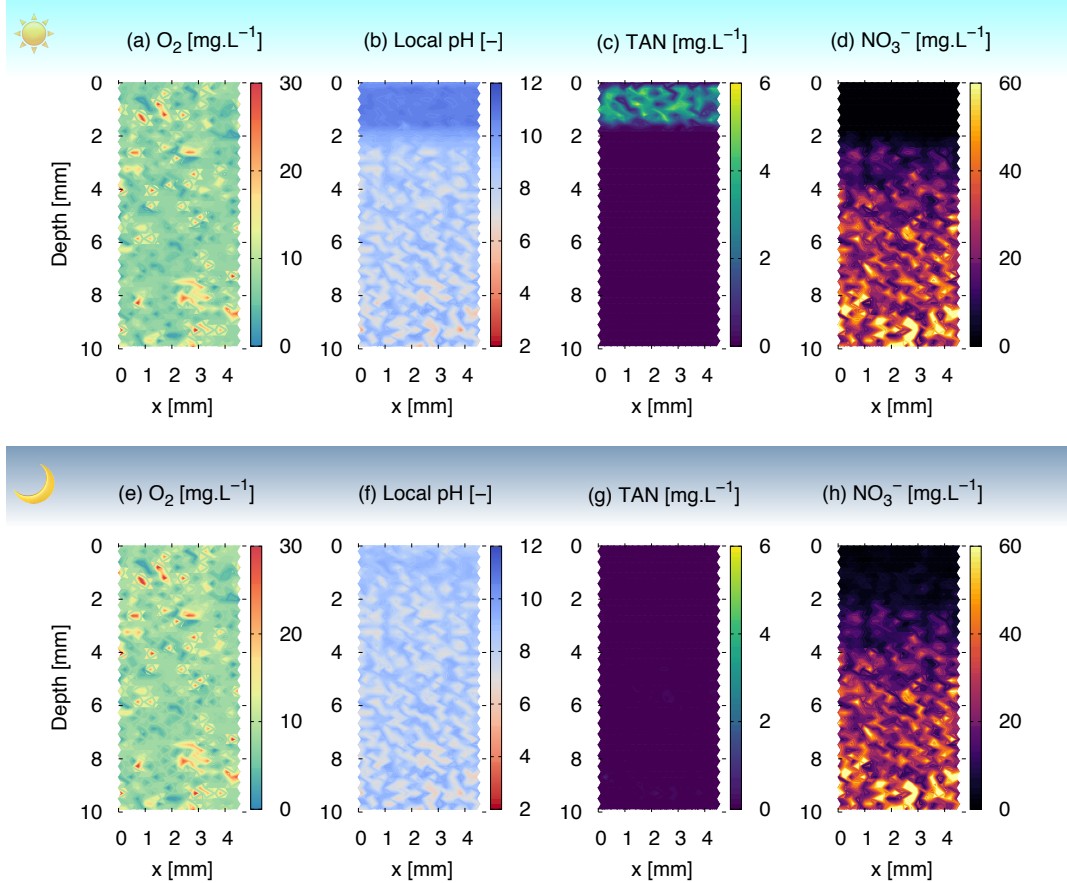

**Figure 4. Diurnal distributions of chemical constituents in the desert biocrust.** A typical result of simulated chemical profile within biocrusts at midday (top panel) and at midnight (bottom panel) at field capacity (wet but unsaturated). (a, e) The profile of dissolved oxygen is relatively stable during the day and night cycle. This implies that gas transport from atmosphere is fast enough to override the consumption/production of microbial community. (b, f) The profile of pH changes in contrast to that of oxygen. During the day, the top of the crust (within 2 mm) exhibits strong alkalisation, marked as blue in the figure. During the night, pH at the top goes back to the similar level as below 2 mm. (c, g) Total ammonia nitrogen (TAN) increases during day on the top of the crust due to microbial production ($N_2$ fixation) and decreases during night by microbial consumption. (d, h) Nitrate distribution shows spatio-temporal diurnal cycles with a tendency of cumulation below 4-5 mm.



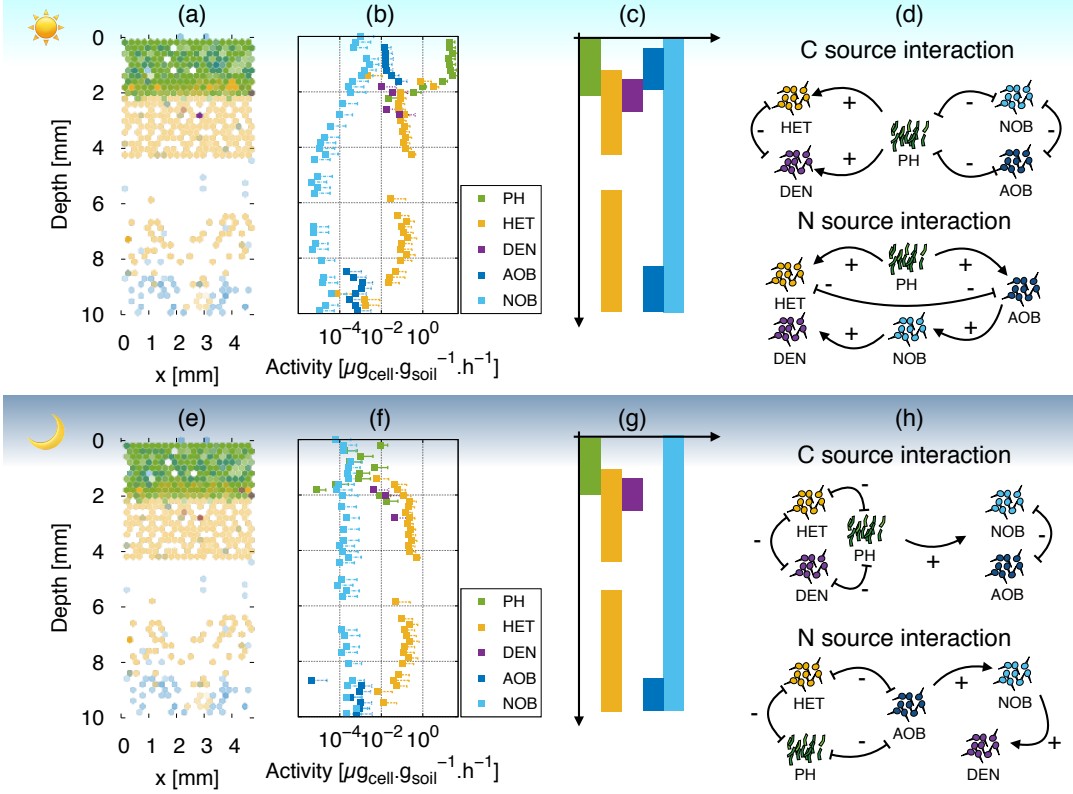

**Figure 5. Diurnal shifts in microbial activity and spatial distributions in desert biocrusts** Simulated biological activity profiles within wet biocrusts at midday (top panel) and at midnight (bottom panel) at field capacity. Microbial activity is expressed in $\mu g_{cell} \cdot g_{soil}^{-1} \cdot h^{-1}$ (product of local biomass and growth rate per gram of soil). (a, e) Spatial distribution of microbial activity is given. Five colours (green, yellow, purple, dark blue, and light blue) represent the microbial groups (photoautotrophs (PH), aerobic heterotrophs (HET), anaerobic heterotrophs (DEN), ammonia oxidisers (AOB), nitrite oxidiser (NOB), respectively). Higher activity is shown with more stronger colour. Vertical distribution of microbial activity at midday (b) and at midnight (f). The spatial distribution of activity is averaged with respect to the horizontal direction. Only upper standard deviations (+1 S.D.) are shown considering the log scale plot. (c, g) The spatial extent of the activity of each functional microbial group within the biocrust is represented by a bar (of the assigned colours above). (d, h) Phototrophic activity changes during day and night, resulting in distinctive trophic interaction patterns over carbon and nitrogen source. (+) and (−) indicate mutualistic and competitive interactions, respectively.



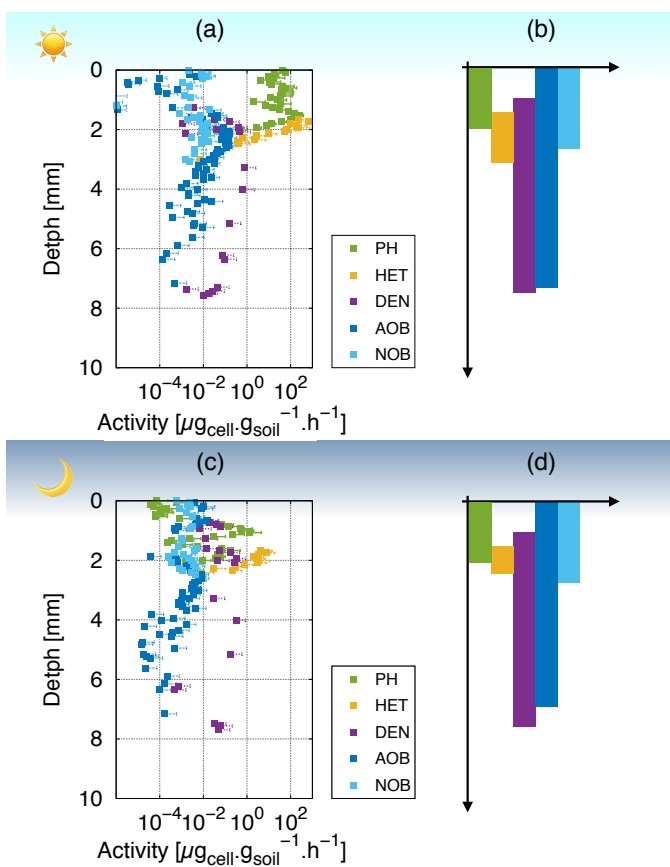

**Figure 6. Diurnal variations in microbial activity within saturated biocrusts.** Simulated microbial activity profiles and vertical stratification at midday (a) and midnight (c). The spatial distribution of microbial activity is averaged with respect to the horizontal direction for 10 independent simulations. (only upper standard deviations (+1 S.D) are shown considering the log scale) (b,d) Based on the vertical distribution of microbial community members, the depth containing the activity of each microbial group within the biocrust is marked by the bars (with respective colour coding).





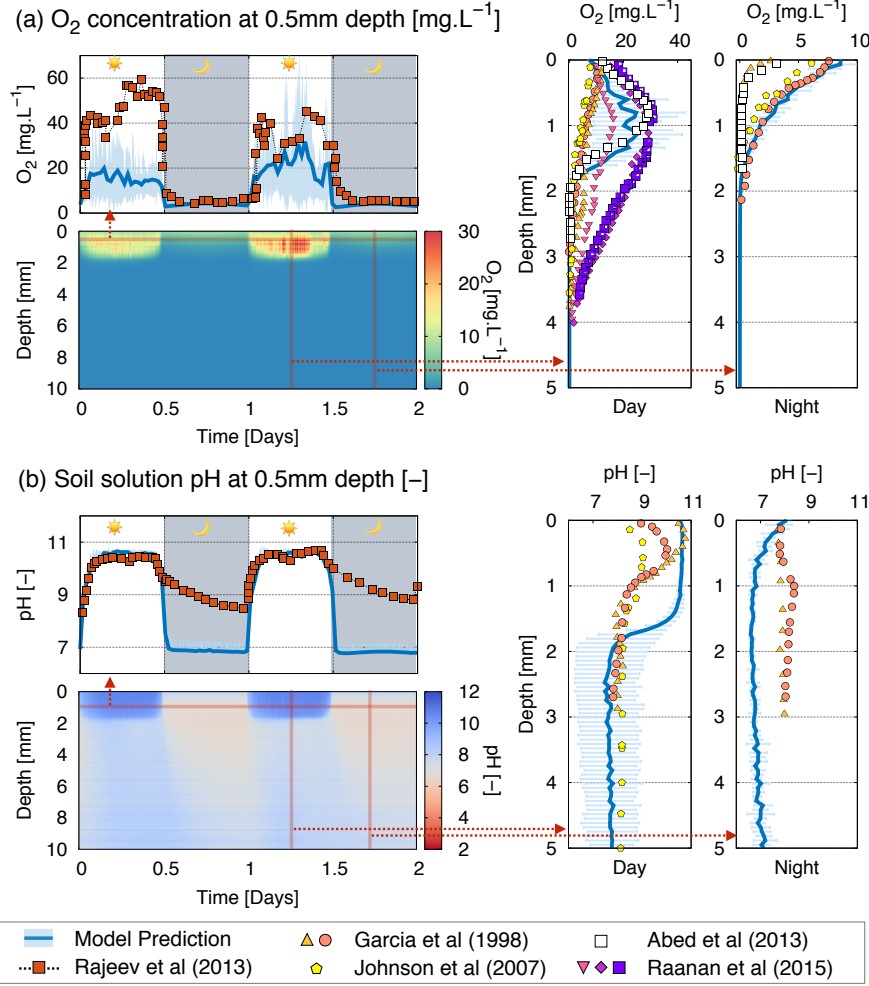

**Figure 7. Oxygen and pH profiles within saturated biocrusts.** Spatio-temporal dynamics of (a) oxygen profile and (b) pH profile of modelled biocrusts (fully saturated) under diurnal cycles. The horizontal average of profiles is taken and ten independent simulations are averaged to see the general dynamics of various biocrusts. For comparisons, 500 μm depth is chosen to represent the temporal behaviour of the top crust. Depth averaged profiles at midday and midnight are used to compare with experimental measurements of biocrust response under light and dark conditions.





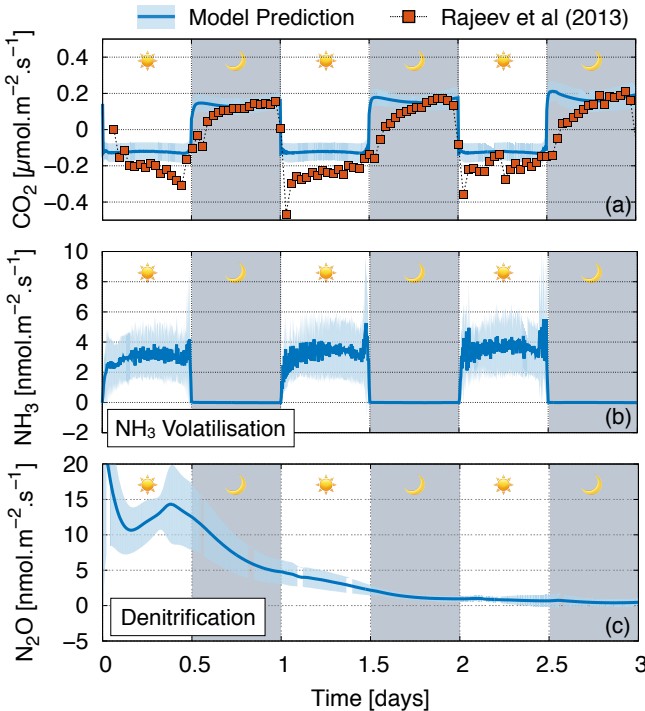

**Figure 8. Gas effluxes from saturated biocrusts.** Gaseous efflux from saturated biocrusts is concomitantly obtained with chemical profiles and microbial activity from 10 independent simulations of the model. (a) $CO_2$ efflux shows diel cycles of uptake (during daytime) and release (during nighttime). The averaged $CO_2$ efflux dynamics are compared with an observation (red squares from Rajeev et al. (2013)). (b) $NH_3$ efflux dynamics show volatilisation of ammonia gas mainly caused by alkalisation of the top crust during daytime, resulting in a net volatilisation rate of about $500\ \mathrm{nmol.m^{-2}day^{-1}}$. (c) $N_2O$ efflux is also calculated as an indicator of denitrification. Observed is that the highest denitrification rate during the first 1-2 days.