# Peer review of "Hydration Status and Diurnal Trophic Interactions Shape Microbial Community Function in Desert Biocrusts"

_Biogeosciences, 2017_

## Referee Comment (RC1) · Anonymous Referee #1 · 20 Jul 2017

In their manuscript, Kim and Or introduce a mechanistic biocrust model, which describes the functioning of biocrust communities with a special emphasis on carbon and nitrogen cycling within these systems. This is a highly relevant topic and the authors made large efforts to include many factors relevant in biocrusts. The results look logic and reasonable and will be relevant for many aspects of biocrust research. Nevertheless, there are some aspects of biocrust functioning, as e.g. leaching of nutrients, erosion processes and more complex N cycling mechanisms, which were not considered in the model and which at least should be mentioned/discussed in the discussion section. More detailed information on this is given below. Detailed comments: Page 2, line 10 ff.: "The settlement of photoautotrophic organisms is followed by other phototrophic, heterotrophic and chemoautotrophic microorganisms". Here, the publication by Pepe-Ranney et al. may be considered, where settlement of non-cyanobacterial diazotrophic bacteria prior to cyanobacteria is described. C Pepe-Ranney, C Koechli, R Potrafka, C Andam, E Eggleston (2016) Non-cyanobacterial diazotrophs mediate dinitrogen fixation in biological soil crusts during early crust formation. The ISME journal 10 (2), 287-298. Page 2, line 26 ff.: Here, you should consider introducing the work of Porada and co-authors, where lichens and mosses as important biocrust compounds have been modelled, e.g.: Porada, P., Weber, B., Elbert, W., Pöschl, U. & Kleidon, A. (2013) Estimating global carbon uptake by lichens and bryophytes with a process-based model. Biogeosciences 10: 6989-7033. Porada, P., Weber, B., Elbert, W., Pöschl, U. & Kleidon, A. (2014) Estimating impacts of lichens and bryophytes on global biogeochemical cycles. Global Biochemical Cycles 28: 71-85. Porada, P., Lenton, T.M., Pohl, A., Weber, B., Mander, L., Donnadieu, Y., Beer, C., Pöschl, U., Kleidon, A. (2016) Strong weathering and climate effects of early lichens and bryophytes in the Late Ordovician. Nature Communications 7: 12113. Porada, P., Pöschl, U., Kleidon, A., Beer, C., Weber, B. (2017) Estimating global nitrous oxide emissions by lichens and bryophytes with a process-based productivity model. Biogeosciences 14: 1593-1602 Page 2, line 15 ff. and Fig. 1: The uppermost millimeters of a biocrust clearly differ from the rest, as e.g. the soil texture is finer etc.. It seems to me that this has not been considered, but on the other hand, the uppermost millimeter or two are shown in a different color. What does the different color imply? Please clarify! Page 5, line 18 ff.: Gas diffusion is most probably affected by finer soil at the surface and by extracellular polysaccharides (EPS). This at least needs to be discussed in a proper way as a potential source of error. Page 7, equation R3: The minus in $CO_3^{2-}$ needs to be written in superscript Page 8, line 15: As I understand from the later text, diazotrophic photoautotrophs, i.e. cyanobacteria capable of fixing atmospheric nitrogen are forming the first group. This has to be made clear. Page 14, line 21 ff.: The oxygen profile shows no major variation with depth or time of the day. I am quite sure that this does not reflect natural conditions in biocrusts under unsaturated water conditions, but that higher

oxygen contents occur in the uppermost millimeters of a biocrust during daytime. As stated above, multiple scientific results have shown that 1. Soil texture of biocrusts is much finer in the uppermost layer, and 2. There is a dense layer of EPS in the uppermost millimeters, both causing a constrained exchange of gases with the atmosphere. This definitely has to be discussed in an appropriate manner and should be improved in a follow-up version of the model. Page 14, line 24 ff.: It is hard to recognize a diurnal cycle in the nitrate profile in Fig. 4, as stated by the authors. This has to be clarified. It also does not become clear, why a nitrate accumulation below 4-5 mm should be caused by inhibited denitrification. One normally would expect this inhibition to take place at shallower depth, also due to the occurrence of oxygen. Please clarify! Page 15, line 7 ff.: Some things are striking in the distribution of organism groups along the profile and need explanations. First, there are only very few denitrifiers present and they occur at only ∼2-3 mm depth. This looks strange, as one would expect more of them occurring at larger depth. Second, there is a pronounced organism gap at ∼4-6 mm depth. Can you please give explanations for these features. Page 15, line 32 ff.: Although the simulation has been made for fully saturated water conditions, which indeed only rarely occur within biocrusts under natural conditions, I still expect it to more closely reflect the actual distribution and activity patterns in biocrusts with a dense surface especially contraining the gaseous surface exchange.

Minor issues: Page 1, Line 9: assemblies instead of assembly Page 1, Line 12: remove "the" Page 1, Line 18: remove comma Page 1, Line 20: include "the" -> protects the soil surface Page 3, line 22: Matrix potential instead of matric potential Page 12, line 13 ff.: This sentence is incomplete and needs to be corrected Page 12, line 26 ff.: Formatting of this sentence has to be corrected with regard to brackets. Errors like that occur also in other parts of the manuscript and need to be corrected. There are many, many more of these minor language issues. Thus, the manuscript needs to be thoroughly checked and corrected by a native speaker.

---

## Referee Comment (RC2) · Anonymous Referee #2 · 23 Jul 2017

The submitted manuscript by Kim and Or titled "Hydration status and diurnal trophic interactions shape microbial community function in desert biocrusts" builds a mechanistic model to look biological, physical and chemical process under different environmental conditions (e.g., temperature, light and hydrations status) and their interactions. In addition, each of these processes and conditions was simulated under different spatial (e.g., soil column) and temporal (e.g., diurnal cycles) resolutions. Overall the study has put a great deal of effort to create a detailed model that captures the high variability found within a biocrust system. It is also apparent that the authors were thoughtful in the specific metrics they chose to include, and why others, while relevant were omitted. However, with this detail it appears complicated for the reader when identifying what

was tested where and how it pertained to the results. I suggest a general conceptual diagram that ties the different biotic and abiotic variables simulated and how they are each related to help guide the reader to understand what was done and why.

GENERAL COMMENTS:

The abstract does not adequately demonstrate the results and conclusions of the study. By stating "the model captures key features of observed microbial activity and distribution. . ." and "new insights into the highly dynamic localized processes that shape biocrust functioning. . ." but don't actually state what those features and functions are the reader is unable to grasp the main conclusions of the study and are left without much to work with.

At the end of the introduction the authors state the organization of the manuscript. I think this is very useful and recommend expanding this further into a table that states the specific variables within the model and their different sub-variables. For instance there could be a biological header and then the sub-headers could be the different main biological variables utilized. Then a chemical primary header and then perhaps an abiotic header with those variables underneath. Whatever format the authors choose, I think having a concise table of what is tested and what the general output would be could really help the reader. Additionally, by having this table the reader can refer to the equations presented in the text and see where they fit into the model in general. That said, if a robust conceptual model is included to better understand the interrelatedness of the variables this table could either strengthen the organization, or perhaps duplicate it. I strongly feel a conceptual model would be useful, however defer to the authors if they want to present it in table form or some other visual, or both. The discussion does not follow a similar organization to the results. Having the uniquely different headers makes it difficult to return to where these findings (and discussions) were reported in the results. Where possible, I suggest having the results and discussion headers more closely follow one another.

MINOR COMMENTS:

Pg 1 Ln 13: Remove "for" to read ". . . carbon covering over 70% of land. . ."

Pg 1 Ln 17: An appropriate citation to be added: Rodríguez-Caballero, E., M. Á. Aguilar, Y. C. Castilla, S. Chamizo, and F. J. Aguilar. 2015. Swelling of biocrusts upon wetting induces changes in surface micro-topography. Soil Biology and Biochemistry 82:07–111

Pg 1 Ln 18: In addition to Chamizo et al. 2012, a newer citation to be added: Faist, A.M., Herrick, J.E., Belnap, J., Van Zee, J.W. and Barger, N.N., 2017. Biological soil crust and disturbance controls on surface hydrology in a semi‐arid ecosystem. Ecosphere, 8: e01691

Pg 2 Ln 13: In the later stages of succession the cyanobacteria are not necessarily "replaced" by other photoautotrophs as they remain in high abundance well into the late successional phases. I would remove this statement.

Pg 2 Ln 28: The word "sketchy" does not feel appropriate for this context. Replace with something more universal such as ". . .sensitive ecosystem remain unclear."

Pg 2 Ln 28: The sentence starting with "Many field and laboratory studies. . ." does not make sense. Don't all studies rely on statistical analyses of the results to deduce impacts? Please reword or clarify.

Pg 13 Ln 14: I don't think "ingredients" is the best term to use. Perhaps "components"

Pg 14 Ln 8-9: Because many of these cyanobacteria form sheaths that they can move up and down the soil column their abundance across a biocrust depth can vary depending on the light. Are you stating here that they decreased as you go down the column? I think a quick explanation how their movement across the soil column could warrant the projected organization would be helpful.

Pg 14 Ln 25: I would add "occurred" after denitrification.

Pg 14: Ln 30: What do you mean by internal trophic interactions? Interspecies? Intraspecies? Within the community?

Pg 14 Ln 29- Pg 15 Ln 6: These would probably fit better in the microbial methods section as opposed to the results section as they are descriptors of what you calculated rather than the actual findings of what you calculated.

Pg 15 Ln 13: state what figure number when say "green in figure" and the same recommendation goes for the rest of the text, when referring to a figure state the specific figure of reference.

Pg 16 Ln 1- 16: I really like the comparison of the simulated data with that of real world observations. However, the Garcia-Pichel and Belnap 1996 reference doesn't match the Garcia et al 1998 citation. Are these different studies? If so, I would site the Garcia-Pichel et al. 1998 in the text.

Pg 16 Ln 11: Change from "quantitatively" to "quantitative"

Pg 20 Ln 14: change "an hydrated" to "a hydrated"

―――――――――――――――

---

## Author Comment (AC2) · 30 Aug 2017

**Response to Referees' Comments : bg-2017-157**

Hydration status and diurnal trophic interactions shape microbial community function in desert biocrusts

Minsu Kim[1] and Dani Or[1]

[1] Soil and Terrestrial Environmental Physics (STEP),

Department of Environmental Systems Sciences (USYS),

ETH Zürich, 8092 Zürich, Switzerland

We thank the reviewer for the constructive comments and suggestions for the manuscript. In the following we provide a point-by-point response to all comments.

The submitted manuscript by Kim and Or titled Hydration status and diurnal trophic interactions shape microbial community function in desert biocrusts builds a mechanistic model to look biological, physical and chemical process under different environmental conditions (e.g., temperature, light and hydrations status) and their interactions. In addition, each of these processes and conditions was simulated under different spatial (e.g., soil column) and temporal (e.g., diurnal cycles) resolutions. Overall the study has put a great deal of effort to create a detailed model that captures the high variability found within a biocrust system. It is also apparent that the authors were thoughtful in the specific metrics they chose to include, and why others, while relevant were omitted. However, with this detail it appears complicated for the reader when identifying what was tested where and how it pertained to the results. I suggest a general conceptual diagram that ties the different biotic and abiotic variables simulated and how they are each related to help guide the reader to understand what was done and why.

We thank the reviewer for the encouraging comments and the constructive suggestions to improve the manuscript. Following the suggestion, we will introduce a conceptual diagram of the current work for readers. A draft for the conceptual diagram is shown in Fig. 1 in this letter. The figure summarises the processes we consider in the model together with variables and parameters involved.

GENERAL COMMENTS:
The abstract does not adequately demonstrate the results and conclusions of the study. By stat-

[Figure]

Figure 1: A conceptual diagram of the desert biocrust model (DBM) in this work.

ing "the model captures key features of observed microbial activity and distribution ..." and "new insights into the highly dynamic localized processes that shape biocrust functioning..." but don't actually state what those features and functions are the reader is unable to grasp the main conclusions of the study and are left without much to work with.

We will revise and expand for clarity as suggested by the reviewer.

At the end of the introduction the authors state the organization of the manuscript. I think this is very useful and recommend expanding this further into a table that states the specific variables within the model and their different sub-variables. For instance, there could be a biological header and then the sub-headers could be the different main biological variables utilized. Then a chemical primary header and then perhaps an abiotic header with those variables underneath. Whatever format the authors choose, I think having a concise table of what is tested and what the general output would be could really help the reader. Additionally, by having this table the reader can refer to the equations presented in the text and see where they fit into the model in general. That said, if a robust conceptual model is included to better understand the interrelatedness of the variables this table could either strengthen the organization, or perhaps

duplicate it. I strongly feel a conceptual model would be useful, however defer to the authors if they want to present it in table form or some other visual, or both. The discussion does not follow a similar organization to the results. Having the uniquely different headers makes it difficult to return to where these findings (and discussions) were reported in the results. Where possible, I suggest having the results and discussion headers more closely follow one another.

We appreciate the detailed suggestions about the organisation of the manuscript. We will work on the manuscript together with the conceptual diagram (Fig. 1 in this letter, to be added in the revised manuscript) so it reads better in the revised manuscript.

MINOR COMMENTS:

- Pg 1 Ln 13: Remove "for" to read "... carbon covering over 70% of land..."

    – amended.

- Pg 1 Ln 17: An appropriate citation to be added: Rodrguez-Caballero, E., M. . Aguilar, Y. C. Castilla, S. Chamizo, and F. J. Aguilar. 2015. Swelling of biocrusts upon wetting induces changes in surface micro-topography. Soil Biology and Biochemistry 82:07111

    – The reference will be added.

- Pg 1 Ln 18: In addition to Chamizo et al. 2012, a newer citation to be added: Faist, A.M., Herrick, J.E., Belnap, J., Van Zee, J.W. and Barger, N.N., 2017. Biological soil crust and disturbance controls on surface hydrology in a semi arid ecosystem. Ecosphere, 8: e01691

    – We thank for introducing us a recent work! We will add the reference.

- Pg 2 Ln 13: In the later stages of succession the cyanobacteria are not necessarily "replaced" by other photoautotrophs as they remain in high abundance well into the late successional phases. I would remove this statement.

    – The sentence has been removed.

- Pg 2 Ln 28: The word "sketchy" does not feel appropriate for this context. Replace with something more universal such as " ...sensitive ecosystem remain unclear."

    – amended.

- Pg 2 Ln 28: The sentence starting with "Many field and laboratory studies..." does not make sense. Dont all studies rely on statistical analyses of the results to deduce impacts? Please reword or clarify.

  – We apologise for the unclear statement. We meant that there is a lack of mechanistic models of biocrusts. The sentence will be rewritten.

- Pg 13 Ln 14: I dont think "ingredients" is the best term to use. Perhaps "components"

  – amended.

- Pg 14 Ln 8-9: Because many of these cyanobacteria form sheaths that they can move up and down the soil column their abundance across a biocrust depth can vary depending on the light. Are you stating here that they decreased as you go down the column? I think a quick explanation how their movement across the soil column could warrant the projected organization would be helpful.

  – In the model, phototrophs were simply inoculated over the depth to follow light intensity with an exponentially decaying function. We note that this is only the initial condition and their distribution and abundance change during simulations following diurnal cycles until it reaches to a quasi-steady state.

- Pg 14 Ln 25: I would add occurred after denitrification.

  – amended.

- Pg 14: Ln 30: What do you mean by internal trophic interactions? Interspecies? Intraspecies? Within the community?

  – "internal trophic interactions" is rewritten as "trophic interactions within the biocrust community".

- Pg 14 Ln 29 - Pg 15 Ln 6: These would probably fit better in the microbial methods section as opposed to the results section as they are descriptors of what you calculated rather than the actual findings of what you calculated.

  – We will move this part to the method section.

- Pg 15 Ln 13: state what figure number when say "green in figure" and the same recommendation goes for the rest of the text, when referring to a figure state the specific figure of reference.

- – amended.

- Pg 16 Ln 1- 16: I really like the comparison of the simulated data with that of real world observations. However, the Garcia-Pichel and Belnap 1996 reference doesnt match the Garcia et al 1998 citation. Are these different studies? If so, I would site the Garcia-Pichel et al. 1998 in the text.

  - – The citation in Fig. 7 was incorrect and the legend (not the text) will be corrected as Garcia-Pichel and Belnap (1996).

- Pg 16 Ln 11: Change from "quantitatively" to "quantitative"

  - – amended.

- Pg 20 Ln 14: change "an hydrated" to "a hydrated"

  - – amended.

---

## Editor Comment (EC1) · A. Antoninka (Editor) · 15 Sep 2017

Please submit the revised manuscript as discussed in your response to reviewers 1 and 2.

---

## Author Response (AR1)

**Response to Referees' Comments : bg-2017-157**

Hydration status and diurnal trophic interactions shape microbial community function in desert biocrusts

Minsu Kim[1] and Dani Or[1]

[1] Soil and Terrestrial Environmental Physics (STEP),

Department of Environmental Systems Sciences (USYS),

ETH Zürich, 8092 Zürich, Switzerland

Dear Editor,

We thank you and the both reviewers for the constructive comments and suggestions. In the following we provide a point-by-point response to all comments. A marked-up manuscript version is attached at the end of the response. Most of comments are identical to the author comment in the discussion panel and changes that have been made are indicated with (1) line numbers in the marked-up version and (2) new line numbers in the final version of the manuscript.

**Review 1**

In their manuscript, Kim and Or introduce a mechanistic biocrust model, which describes the functioning of biocrust communities with a special emphasis on carbon and nitrogen cycling within these systems. This is a highly relevant topic and the authors made large efforts to include many factors relevant in biocrusts. The results look logic and reasonable and will be relevant for many aspects of biocrust research. Nevertheless, there are some aspects of biocrust functioning, as e.g. leaching of nutrients, erosion processes and more complex N cycling mechanisms, which were not considered in the model and which at least should be mentioned/discussed in the discussion section.

We thank the reviewer for the kind and encouraging comments. Indeed the present biocrust model is a very reduced version of real biocrusts with a focus on microbial interactions in space and time using a few external variables. We included a discussion that places the proposed model in context of more general processes that were not included here to highlight limitations and complexity of these unique ecosystems (page 22, line 3-11; new page 21, new line 24-32).

Page 2, line 10 ff.: The settlement of photoautotrophic organisms is followed by other phototrophic, heterotrophic and chemoautotrophic microorganisms. Here, the publication by Pepe-Ranney et al. may be considered, where settlement of non-cyanobacterial diazotrophic bacteria prior to cyanobacteria is described.

Following the suggestion, the reference has been added (page 2, line 15; new page 2, new line 13).

Page 2, line 26 ff.: Here, you should consider introducing the work of Porada and co-authors, where lichens and mosses as important biocrust compounds have been modelled.

We thank the reviewer for bringing this interesting work to our attention. We have stated in the original manuscript that the proposed model focuses on the early stages of biocrust formation prior to establishment of lichens and mosses. Nevertheless, for completeness, we will add this reference (page 2, line 34; new page 2, new line 30-32).

Fig. 1: The uppermost millimeters of a biocrust clearly differ from the rest, as e.g. the soil texture is finer etc.. It seems to me that this has not been considered, but on the other hand, the uppermost millimeter or two are shown in a different color. What does the different color imply? Please clarify

We apologise for the confusion in the representation. A different colour was introduced to the upper most region (up to around 5 mm) to indicate a region of interest in this work (see Figure 7). It is correct that the pre-assigned physical structure was statistically identical to the below crust (A finer soil texture is not assigned at the surface, i.e. same roughness parameters and porosity were used.). The caption is rewritten accordingly (page 30; new page 29).

Page 5, line 18 ff.: Gas diffusion is most probably affected by finer soil at the surface and by extracellular polysaccharides (EPS). This at least needs to be discussed in a proper way as a potential source of error.

We thank for your constructive comments. We certainly agree that these factors greatly modify gas diffusion and water holding capacity across biocrusts. The possible modifications by finer soil texture at the surface and by EPS are discussed in the revised manuscript (page 19, line 31; new page 19, new line 22-30).

Page 7, equation R3: The minus in CO32- needs to be written in superscript.

This has been amended (page 7, equation (R3); new page 7, equation (R3)).

 As I understand from the later text, diazotrophic photoautotrophs, i.e. cyanobacteria capable of fixing atmospheric nitrogen are forming the first group. This has to be made clear.

The first group is now described as diazotrophic photoautotrophs (page 9, line 12-13; new page 9, new line 12).

 The oxygen profile shows no major variation with depth or time of the day. I am quite sure that this does not reflect natural conditions in biocrusts under unsaturated water conditions, but that higher oxygen contents occur in the uppermost millimeters of a biocrust during daytime. As stated above, multiple scientific results have shown that 1. Soil texture of biocrusts is much finer in the uppermost layer, and 2. There is a dense layer of EPS in the uppermost millimeters, both causing a constrained exchange of gases with the atmosphere. This definitely has to be discussed in an appropriate manner and should be improved in a follow-up version of the model.

The profile of dissolved oxygen was appeared relatively stable due to the mass transfer between gas and liquid in unsaturated soils which is assumed to be very rapid (instantaneous equilibration by Henry's law, see section 2.3.2). This assumption is reasonable because of large liquid-vapour interfacial area and thin water film thickness of unsaturated soils. As the reviewer pointed out, we agree that biocrusts loaded with EPS and finer texture at the surface would affect the exchange of gases. These factors can retard the mass transfer between soil water and the atmosphere by decreasing interfacial area at a relatively wet condition (finer soil texture) and by sustaining thick water film thickness after wetting (swelling of EPS or increase in water holding capacity).

In terms of the current modelling approach, assigning a finer soil texture is possible by using a low porosity or a higher fractal dimension on the uppermost part of biocrusts. As an extended model of biocrusts, local EPS contents can be related to waterfilm thickness at a given matric potential (This would be interesting to investigate the role of EPS in wet-dry cycles that was not included in this work.). Overall, these factors change the water retention behaviour at the surface and the gas percolation that changes boundary conditions for Henry's law. This implies gas diffusion can be hindered even under unsaturated biocrusts at a certain range of hydration conditions as the reviewer mentioned. We greatly appreciate the comment and this aspect has been discussed in the revised manuscript as we mentioned above (page 19, line 31; new page 19, new line 22-30).

 It is hard to recognize a diurnal cycle in the nitrate profile in Fig. 4, as stated by the authors. This has to be clarified. It also does not become clear, why a nitrate accumulation below 4-5 mm should be caused by inhibited denitrification. One normally would expect this inhibition to take place at shallower depth, also due to the occurrence of oxygen. Please clarify!

We thank the reviewer for the insightful comment, and recognise issues with the distribution of nitrate. Figure 4 was meant to show that the nitrate profile does NOT exhibit clear diurnal patterns which differ from the nitrate profile under saturated biocrusts (See Figure S4 in Supplementary text). The accumulation of nitrate below 4-5 mm was clearly due to the lack of denitrification and the activity of nitrite oxidising bacteria producing nitrate (see Figure 5, denitrification (purple) only occurs locally at the shallow depth while NOB (light blue) appears through out the depth.). We attribute this local denitrification to the creation of local anoxic region, where aerobic organisms scavenge oxygen at the certain region that is disconnected from the percolating cluster of gas phase. High abundance of aerobic organisms (HET, AOB, NOB) yield these local anoxic regions above 4 mm depth (These were already described in the manuscript, new page 16, new line 15-16 and new page 20 new line 5-7). The incorrect descriptions are revised (page 15, line 28-29 and Fig. 4 in page 33; new page 15, new line 28-39 and Fig. 4 in new page 32).

 Some things are striking in the distribution of organism groups along the profile and need explanations. First, there are only very few denitrifiers present and they occur at only 2-3 mm depth. This looks strange, as one would expect more of them occurring at larger depth. Second, there is a pronounced organism gap at 4-6 mm depth. Can you please give explanations for these features.

As we stated above, under unsaturated conditions, denitrification occurs at local anoxic region supported by oxygen scavenging aerobic organisms at patches that are disconnected from the atmosphere. This is not necessarily a 1-D stratified structure and may occur as hot spots for denitrification in this context. The absence of denitrification at larger depth was due to the inhibition by oxygen. In the model, the low activity of aerobic organisms cannot create local anoxic region because of the fast gas transport from the atmosphere overriding their consumptions (i.e., instantaneous equilibration of Henry's law). Regarding the second point, we note that Fig. 5 is a typical result of the DBM simulation (not an averaged result from multiple simulations like Fig. 6). We chose to show this way to put an emphasis on the spatial variability even within at the micro-scale (such as the arise of local anoxic region) under unsaturated conditions. However,

due to the light penetration, the vertical stratification was still observed from all the simulations in a similar manner. The microbial activity below around 4 mm is anyway sparse and the gap of activity is domain-specific (randomly created spatial heterogeneity of the physical soil domain). We would like to focus on the high abundance of organisms above 4 mm that are supported by diffusion of fixed carbon and nitrogen at the surface by diazotrophic phototrophs. We clarified this in the revised manuscript (page 16, line 3-4 and Fig. 5 in page 34; new page 15, new line 33 and Fig. 5 in new page 33).

Page 15, line 32 ff.: Although the simulation has been made for fully saturated water conditions, which indeed only rarely occur within biocrusts under natural conditions, I still expect it to more closely reflect the actual distribution and activity patterns in biocrusts with a dense surface especially contraining the gaseous surface exchange.

We certainly agree that we gain valuable knowledge about microbial distributions and activities within biocrusts under fully saturated conditions. However, these conditions (especially, soils fully immersed in water) are rarely encountered in desert biocrusts even if fine soil texture and EPS can prolong such conditions. Therefore, understanding biocrusts under unsaturated or dynamic hydration (wetting-drying) is important to quantify microbial interactions and their role for carbon and nitrogen cycling. We are aware of that there is a limit to directly observe microbial activities under unsaturated soils. This modelling work attempts to predict physical and chemical micro-environments to figure out how microbial trophic interactions can be influenced by saturation degree. For instance, aerobic heterotrophs and nitrifiers that would benefit within moderately unsaturated soils thanks to oxygen transport from gas phase. On the other hand, fully saturated conditions favour anaerobic processes, such as denitrification. This implies that studies on fully saturated biocrusts might underestimate or overestimate certain aspects of biocrusts for its role for carbon and nitrogen cycling under natural conditions.

MINIOR ISSUES:

- Page 1, Line 9: assemblies instead of assembly Page 1, Line 12: remove "the" Page 1, Line 18: remove comma Page 1, Line 20: include "the" → protects the soil surface

    - Amended.

- Page 3, line 22: Matrix potential instead of matric potential

    - To be consistent with other modelling papers published along with this work, we

decided to keep the term matric potential instead of matrix potential.

- Page 12, line 13 ff.: This sentence is incomplete and needs to be corrected

  – The sentence is corrected (page 13, line 11-12; new page 13, new line 11-12).

- Page 12, line 26 ff.: Formatting of this sentence has to be corrected with regard to brackets. Errors like that occur also in other parts of the manuscript and need to be corrected. There are many, many more of these minor language issues. Thus, the manuscript needs to be thoroughly checked and corrected by a native speaker.

  – We have made every effort to correct the language issues in the revised manuscript. You can find the corrections in the marked-up manuscript.

**Review 2**

The submitted manuscript by Kim and Or titled Hydration status and diurnal trophic interactions shape microbial community function in desert biocrusts builds a mechanistic model to look biological, physical and chemical process under different environmental conditions (e.g., temperature, light and hydrations status) and their interactions. In addition, each of these processes and conditions was simulated under different spatial (e.g., soil column) and temporal (e.g., diurnal cycles) resolutions. Overall the study has put a great deal of effort to create a detailed model that captures the high variability found within a biocrust system. It is also apparent that the authors were thoughtful in the specific metrics they chose to include, and why others, while relevant were omitted. However, with this detail it appears complicated for the reader when identifying what was tested where and how it pertained to the results. I suggest a general conceptual diagram that ties the different biotic and abiotic variables simulated and how they are each related to help guide the reader to understand what was done and why.

We thank the reviewer for the encouraging comments and the constructive suggestions to improve the manuscript. Following the suggestion, we have introduced a table summarising the current model (Table 1, new page 3).

GENERAL COMMENTS:
The abstract does not adequately demonstrate the results and conclusions of the study. By stating "the model captures key features of observed microbial activity and distribution ..." and "new insights into the highly dynamic localized processes that shape biocrust functioning..." but don't actually state what those features and functions are the reader is unable to grasp the main conclusions of the study and are left without much to work with.

We revised the abstract for clarity as suggested by the reviewer (page 1; new page 1).

At the end of the introduction the authors state the organization of the manuscript. I think this is very useful and recommend expanding this further into a table that states the specific variables within the model and their different sub-variables. For instance, there could be a biological header and then the sub-headers could be the different main biological variables utilized. Then a chemical primary header and then perhaps an abiotic header with those variables underneath. Whatever format the authors choose, I think having a concise table of what is tested and what the general output would be could really help the reader. Additionally, by having this table the reader can refer to the equations presented in the text and see where they fit into the

model in general. That said, if a robust conceptual model is included to better understand the interrelatedness of the variables this table could either strengthen the organization, or perhaps duplicate it. I strongly feel a conceptual model would be useful, however defer to the authors if they want to present it in table form or some other visual, or both. The discussion does not follow a similar organization to the results. Having the uniquely different headers makes it difficult to return to where these findings (and discussions) were reported in the results. Where possible, I suggest having the results and discussion headers more closely follow one another.

Following the suggestions, we have included a table summarising variables and parameters used in the current work (page 3, line 11-12; new page 3, new line 6-7).

MINOR COMMENTS:

- Pg 1 Ln 13: Remove "for" to read "... carbon covering over 70% of land..."

  – amended.

- Pg 1 Ln 17: An appropriate citation to be added: Rodrguez-Caballero, E., M. . Aguilar, Y. C. Castilla, S. Chamizo, and F. J. Aguilar. 2015. Swelling of biocrusts upon wetting induces changes in surface micro-topography. Soil Biology and Biochemistry 82:07111

  – The reference has been added (page 1, line 18; new page 1, new line 17-18).

- Pg 1 Ln 18: In addition to Chamizo et al. 2012, a newer citation to be added: Faist, A.M., Herrick, J.E., Belnap, J., Van Zee, J.W. and Barger, N.N., 2017. Biological soil crust and disturbance controls on surface hydrology in a semi arid ecosystem. Ecosphere, 8: e01691

  – We thank for introducing us a recent work! We have added the reference (page 1, line 20; new page 1, new line 19-20).

- Pg 2 Ln 13: In the later stages of succession the cyanobacteria are not necessarily "replaced" by other photoautotrophs as they remain in high abundance well into the late successional phases. I would remove this statement.

  – The sentence has been removed (page 2, line 14-15; new page 2, new line 11-13).

- Pg 2 Ln 28: The word "sketchy" does not feel appropriate for this context. Replace with something more universal such as " ...sensitive ecosystem remain unclear."

  – amended (page 2, line 29-31; new page 2, new line 26-28).

- Pg 2 Ln 28: The sentence starting with "Many field and laboratory studies..." does not make sense. Dont all studies rely on statistical analyses of the results to deduce impacts? Please reword or clarify.

    – We apologise for the unclear statement. We meant that there is a lack of mechanistic models of biocrusts. The sentence is rewritten (page 2, line 31-33; new page 2, new line 28-30).

- Pg 13 Ln 14: I dont think "ingredients" is the best term to use. Perhaps "components"

    – amended.

- Pg 14 Ln 8-9: Because many of these cyanobacteria form sheaths that they can move up and down the soil column their abundance across a biocrust depth can vary depending on the light. Are you stating here that they decreased as you go down the column? I think a quick explanation how their movement across the soil column could warrant the projected organization would be helpful.

    – In the model, phototrophs were simply inoculated over the depth to follow light intensity with an exponentially decaying function. We note that this is only the initial condition and their distribution and abundance change during simulations following diurnal cycles until it reaches to a quasi-steady state.

- Pg 14 Ln 25: I would add occurred after denitrification.

    – amended.

- Pg 14: Ln 30: What do you mean by internal trophic interactions? Interspecies? Intraspecies? Within the community?

    – "internal trophic interactions" is rewritten as "trophic interactions within the biocrust community" (page 16, line 2-3; new page 15, new line 32-33).

- Pg 14 Ln 29 - Pg 15 Ln 6: These would probably fit better in the microbial methods section as opposed to the results section as they are descriptors of what you calculated rather than the actual findings of what you calculated.

    – We moved the part to the method section (page 13, line 18-23; new page 13, new line 17-23).

- Pg 15 Ln 13: state what figure number when say "green in figure" and the same recommendation goes for the rest of the text, when referring to a figure state the specific figure of reference.

    – amended.

- Pg 16 Ln 1- 16: I really like the comparison of the simulated data with that of real world observations. However, the Garcia-Pichel and Belnap 1996 reference doesnt match the Garcia et al 1998 citation. Are these different studies? If so, I would site the Garcia-Pichel et al. 1998 in the text.

    – The citation in Fig. 7 was incorrect and the legend (not the text) will be corrected as Garcia-Pichel and Belnap (1996).

- Pg 16 Ln 11: Change from "quantitatively" to "quantitative"

    – amended.

- Pg 20 Ln 14: change "an hydrated" to "a hydrated"

    – amended.

[revised manuscript text omitted]

**(a)** Physical domain: abstract representation of biocrust cross section

light, temperature, water

Rough surfaces

$O_2$, $CO_2$ — gas exchange

unwrapped surface

abstract geometry

effective film thickness

20 mm

~100μm

Model domain

☐ : gas phase
🟦 : liquid phase
🟫 : solid phase
🔴 : cell and EPS

**(b)** Boundary conditions

Light ☀ — Temperature 🌡

Depth

Wet — Dry

**(c)** Effective film thickness [m] 💧

water — void

Matric potential [–kPa]

**Figure 1. A schematic of physical domain and environmental conditions of the desert biocrust model (DBM).** (a) A cross-section of the physical domain of  desert soil is modelled up to 20 mm and most of microbial activities occur at the uppermost 5 mm, indicating "biocrust", a region of interest in this work (marked in green). The domain comprises hexagonal patches with different physical properties to represent heterogeneity of soil (mimicking soil pores and rough surfaces) with the pre-assigned mean values (See Table S8 in the supplementary text). We note that physical properties of the domain were assigned statistically same for biocrust and below crust regions. The rough surface is simplified with abstract geometries to calculate effective film thicknesses of the surface at patch scale. To represent interference of liquid phase with respect to gas diffusion, we consider two rough soil surfaces (cross-sections) facing each other as described previously (Kim and Or, 2016; Šťovíček et al., 2017). (b) Spatio-temporal variations of light intensity and temperature as boundary conditions (wet and dry) during a diurnal cycle. Surface boundary conditions of temperature changes in accordance with light irradiance during the same period of a day. Unlike light penetration, temperature profile depends on hydration conditions (thermal diffusivity is controlled by the matric potential). Under wetter conditions, thermal diffusivity is higher. (c) Effective thicknesses of water film and void space are determined as a function of matric potential.

[revised manuscript text omitted]

**Figure 7. Oxygen and pH profiles within saturated biocrusts.** Spatio-temporal dynamics of (a) oxygen profile and (b) pH profile of modelled biocrusts (fully saturated) under diurnal cycles. The horizontal average of profiles is taken and ten independent simulations are averaged to see the general dynamics of various biocrusts. For comparisons, 500 μm depth is chosen to represent the temporal behaviour of the top crust. Depth averaged profiles at midday and midnight are used to compare with experimental measurements of biocrust response under light and dark conditions.

[Figure]

**Figure 8. Gas effluxes from saturated biocrusts.** Gaseous efflux from saturated biocrusts is concomitantly obtained with chemical profiles and microbial activity from 10 independent simulations of the model. (a) $CO_2$ efflux shows diel cycles of uptake (during daytime) and release (during nighttime). The averaged $CO_2$ efflux dynamics are compared with an observation (red squares from Rajeev et al. (2013)). (b) $NH_3$ efflux dynamics show volatilisation of ammonia gas mainly caused by alkalisation of the top crust during daytime, resulting in a net volatilisation rate of about $500$ nmol.m$^{-2}$day$^{-1}$. (c) $N_2O$ efflux is also calculated as an indicator of denitrification. Observed is that the highest denitrification rate during the first 1-2 days.